# RopB represses the transcription of *speB* in the absence of SIP in group A *Streptococcus*

Chuan Chiang-Ni[1,2,3,4], Yan-Wen Chen[1], Kai-Lin Chen[3], Jian-Xian Jiang[2], Yong-An Shi[2], Chih-Yun Hsu[1], Yi-Ywan M Chen[1,2,4], Chih-Ho Lai[1,2,4], Cheng-Hsun Chiu[2,4]

**RopB is a quorum-sensing regulator that binds to the SpeB-inducing peptide (SIP) under acidic conditions. SIP is known to be degraded by the endopeptidase PepO, whose transcription is repressed by the CovR/CovS two-component regulatory system. Both SIP-bound RopB (RopB-SIP) and SIP-free RopB (apo-RopB) can bind to the *speB* promoter; however, only RopB-SIP activates *speB* transcription. In this study, we found that the SpeB expression was higher in the *ropB* mutant than in the SIP-inactivated (*SIP\**) mutant. Furthermore, the deletion of *ropB* in the *SIP\** mutant derepressed *speB* expression, suggesting that apo-RopB is a transcriptional repressor of *speB*. Up-regulation of PepO in the *covS* mutant degraded SIP, resulting in the down-regulation of *speB*. We demonstrate that deleting *ropB* in the *covS* mutant derepressed the *speB* expression, suggesting that the *speB* repression in this mutant was mediated not only by PepO-dependent SIP degradation but also by apo-RopB. These findings reveal a crosstalk between the CovR/CovS and RopB-SIP systems and redefine the role of RopB in regulating *speB* expression in group A *Streptococcus*.**

## Introduction

*Streptococcus pyogenes* (group A *Streptococcus*, GAS) is a gram-positive bacterial pathogen that causes various diseases, including pharyngitis, scarlet fever, cellulitis, necrotizing fasciitis, and toxic shock syndrome (Cunningham, 2008). CovR/CovS (control of virulence), previously designated CsrR/CsrS (Levin & Wessels, 1998), is a two-component regulatory system in GAS (Federle et al, 1999). CovS phosphorylates intracellular CovR, and the phosphorylated CovR primarily acts as a transcriptional repressor (Miller et al, 2001; Dalton & Scott, 2004; Gusa et al, 2006; Churchward, 2007). Spontaneous mutations in *covS*, which result in a functional loss in its capacity to phosphorylate CovR, derepress the expression of one

group of virulence factors (streptolysin O, streptokinase, and hyaluronic acid capsule) but repress the transcription of a second group of genes (*speB*, *grab*, and *spd3*) (Sumby et al, 2006; Trevino et al, 2009; Ikebe et al, 2010; Friaes et al, 2015). Specifically, the expression of SpeB protease is down-regulated in the *covS* mutant compared with that in the wild-type strain (Sumby et al, 2006; Trevino et al, 2009; Tran-Winkler et al, 2011; Chiang-Ni et al, 2019a), suggesting that phosphorylated CovR can transcriptionally activate *speB*. Furthermore, Finn et al (Finn et al, 2021) showed that non-phosphorylated CovR can bind to the *speB* promoter and repress *speB* expression. These results suggest that the expression of *speB* is activated by phosphorylated CovR but repressed by non-phosphorylated CovR; however, the deletion of *covR* in the wild-type strain and the *covS* mutant results in the derepression of *speB* (Chiang-Ni et al, 2016). Therefore, the phosphorylated and non-phosphorylated CovR-mediated regulatory mechanisms of *speB* expression require further investigation.

The SpeB cysteine protease is secreted as a zymogen (42 kD), and its protease activity is essential for the autocatalysis of the zymogen to the mature SpeB protease (28 kD) (Doran et al, 1999; Chen et al, 2003). SpeB degrades or cleaves both host proteins (fibrin, fibronectin, vitronectin, immunoglobulins, and complement proteins) and bacterial surface and virulence-associated proteins (Rasmussen & Bjorck, 2002). Therefore, SpeB is considered an important virulence factor, and its expression is tightly regulated in GAS. RopB (Regulator of protease B) is an Rgg-like regulator identified as a transcriptional activator of *speB* (Lyon et al, 1998). Both *speB* and *ropB* are located adjacent to one another on the chromosome but are transcribed in opposite directions (Neely et al, 2003). Two promoters of *speB* are located within the *ropB*–*speB* intergenic region, and the P1 promoter adjacent to *ropB* is the principal promoter for RopB binding and *speB* transcription (Neely et al, 2003). As a quorum-sensing protein, RopB binds to an eight-amino acid leaderless SpeB-inducing peptide (SIP) to induce *speB* expression (Do & Kumaraswami, 2016; Perez-Pascual et al, 2016; Do et al, 2017). Do et al (Do et al, 2019) showed that RopB binds to SIP under acidic conditions, suggesting that SIP mediates the growth

[1]Department of Microbiology and Immunology, College of Medicine, Chang Gung University, Taoyuan, Taiwan   [2]Graduate Institute of Biomedical Sciences, College of Medicine, Chang Gung University, Taoyuan, Taiwan   [3]Department of Orthopedic Surgery, Chang Gung Memorial Hospital at Linkou, Taoyuan, Taiwan   [4]Molecular Infectious Disease Research Center, Chang Gung Memorial Hospital at Linkou, Taoyuan, Taiwan

Correspondence: entchuan@gap.cgu.edu.tw

phase-and pH-dependent *speB* expression. The intracellular SIP concentration is modulated by the endopeptidase PepO. A study showed that the up-regulation of *pepO* in the *covR* mutant mediates SIP degradation, thereby disrupting the RopB-SIP quorum-sensing pathway (Shi et al, 2022). Interestingly, although SIP-bound RopB (RopB-SIP) is required to activate *speB* transcription, RopB-SIP and SIP-free RopB (apo-RopB) have similar DNA-binding activities to the P1 promoter of *speB* (Do et al, 2017). Therefore, the role of apo-RopB in regulating *speB* transcription remains unclear.

RopB is a positive regulator of *speB* and essential for inducing *speB* transcription. In this study, we demonstrate that in the absence of SIP, RopB acts as a transcriptional repressor of *speB*. Therefore, the non-phosphorylated CovR-mediated down-regulation of *speB* in the *covS* mutant is mediated by apo-RopB. These results redefine the current understanding of RopB-mediated regulation of *speB* and reveal a new interaction between the CovR/CovS and RopB-SIP systems in GAS.

# Results

### RopB represses *speB* transcription in the absence of SIP

Our previous study showed that the up-regulation of PepO in the *covR* mutant mediates the degradation of SIP and the down-regulation of *speB* (Shi et al, 2022). The expression of *speB* in the *pepO* mutant was higher than that in the wild-type A20 strain (Fig 1A), suggesting that PepO in the wild-type strain mediates SIP degradation. In this study, we constructed a *pepO* mutant in an SIP-inactivated background to verify the role of PepO in degrading the exogenous supplemented SIP. The start codon of SIP (ATG) in the wild-type A20 strain was substituted with TAG to inactivate SIP translationally. This strain was designated as the *SIP** mutant. The expression of *speB* in the *SIP** mutant was repressed compared with that in the wild-type A20 strain (Fig 1A). The open reading frame of SIP is located in the *ropB–speB* intergenic region (Do et al, 2017). SpeB was up-regulated in the *SIP** mutant complemented with the *ropB–speB* intergenic region (P*SIP*) and *ropB* with its native promoter [P*ropB*(*SIP*+)] compared with that in the vector-control strain (Vec) (Fig 1B), indicating that there are no other undefined factors related to the down-regulation of *speB* in the *SIP** mutant. In the exogenous SIP-supplementation conditions, lower levels of SpeB were observed in the *SIP** mutant compared with its *pepO* isogenic mutant (*SIP**/Δ*pepO*) under the same concentration of SIP treatments (Fig 1C–E), indicating that PepO mediates SIP degradation in the wild-type strain. Furthermore, to verify that the expression of SpeB is induced by RopB under SIP stimuli, the *ropB* gene was deleted in the *SIP** mutant (*SIP**/Δ*ropB*), and the expression of SpeB in this mutant under SIP and the scrambled peptide (SCRA) treatments were analyzed. No difference was observed in SpeB expression in the *SIP**/Δ*ropB* mutant under treatment with 0–1.5 μM SIP (Fig 1F). Less than a 1.2fold increase was found in the RNA level (Fig 1C), suggesting that SIP induces *speB* expression in a RopB-dependent manner.

We also observed that the expression of *speB* was down-regulated in the *ropB* mutant compared with that in the wild-type and *ropB*-complementary strains (Fig 1G), suggesting that

RopB is the transcriptional activator of *speB* (Lyon et al, 1998; Neely et al, 2003). Nonetheless, in comparison with the *SIP** mutant, the deletion of *ropB* in the *SIP** mutant (*SIP**/Δ*ropB*) resulted in the up-regulation of *speB* (Fig 1C and F). Also, the *SIP**/Δ*ropB* mutant (without SIP treatments) showed a significant elevation in SpeB expression compared with that of the *SIP** mutant under the 1.5 μM-SIP treatments (Fig 1F). These results suggest that RopB represses the transcription of *speB* in the *SIP** mutant.

To elucidate the role of RopB in the regulation of *speB* in the presence and absence of SIP, we performed Western blotting and analyzed the levels of SpeB in the wild-type A20, *SIP** mutant, and *ropB* mutant strains. SpeB expression in the *ropB* mutant was down-regulated compared with that in the wild-type A20 strain (Fig 1H), suggesting that RopB positively regulates *speB* transcription in the presence of SIP. Compared with the wild-type strain, SpeB expression was down-regulated in the *SIP** mutant; notably, SpeB expression in the *SIP** mutant was lower than that in the *ropB* isogenic mutant (Fig 1H). Furthermore, the SpeB expression in the *SIP**/Δ*ropB* mutant increased to a level similar to that in the *ropB* isogenic mutant (Fig 1H). Consistent with results from the Western blot analysis, the transcription level of *speB* in the *ropB* isogenic mutant and *SIP**/Δ*ropB* mutant was similar (Fig 1I). In addition, *speB* transcription was down-regulated in the *SIP** mutant compared with that in the *SIP**/Δ*ropB* mutant (Fig 1I). These results suggest that in the SIP-inactivated background, RopB acts as a transcriptional repressor of *speB*.

### RopB inhibits the growth-phase-dependent SpeB expression in the SIP-inactivated *covR* mutant

Compared with that in the wild-type strain, the expression of *speB* was up-regulated in the *covR* and Δ*covR*/Δ*pepO* mutants (Fig 2A), suggesting that CovR also has roles in regulating *speB* expression. To exclude the effects of CovR, the role of RopB in regulating SpeB expression in the presence and absence of SIP was further analyzed in the *covR* mutant. As expected, the deletion of *ropB* in the *covR* mutant (Δ*covR*/Δ*ropB*) down-regulated *speB* transcription in the stationary phase compared with the *covR* mutant (6–7 h, Fig 2B and C). Noticeably, the increase in SpeB expression was still observed in the Δ*covR*/Δ*ropB* mutant after 7 h of incubation (Fig 2B and C), indicating that the growth-phase-dependent SpeB expression was not completely abolished in the absence of RopB. Although the RopB was present, the expression of SpeB both transcriptionally and translationally in the SIP-inactivated *covR* mutant (*SIP**/Δ*covR*) was repressed in comparison to that in the *covR* and the Δ*covR*/Δ*ropB* mutants (Fig 2B and C). To elucidate the role of RopB in regulating SpeB expression in the *SIP**/Δ*covR* mutant, the expression of SpeB in the *SIP**/Δ*covR* mutant and its isogenic *ropB* mutant (*SIP**/Δ*covR*/Δ*ropB*) were compared. The Δ*covR*/Δ*ropB* mutant and *SIP**/Δ*covR*/Δ*ropB* mutant showed a similar level of *speB* transcription after 5 h of incubation (Fig 2D). At the protein level, inactivation of SIP translation in the Δ*covR*/Δ*ropB* mutant (*SIP**/Δ*covR*/Δ*ropB*) had a minor effect on SpeB expression compared with that in the Δ*covR*/Δ*ropB* mutant (Fig 2E), indicating that SIP-mediated SpeB expression is primarily through RopB. Furthermore, we found that SpeB expression was derepressed in the *SIP**/Δ*covR*/Δ*ropB* mutant compared with that in the *SIP**/Δ*covR*

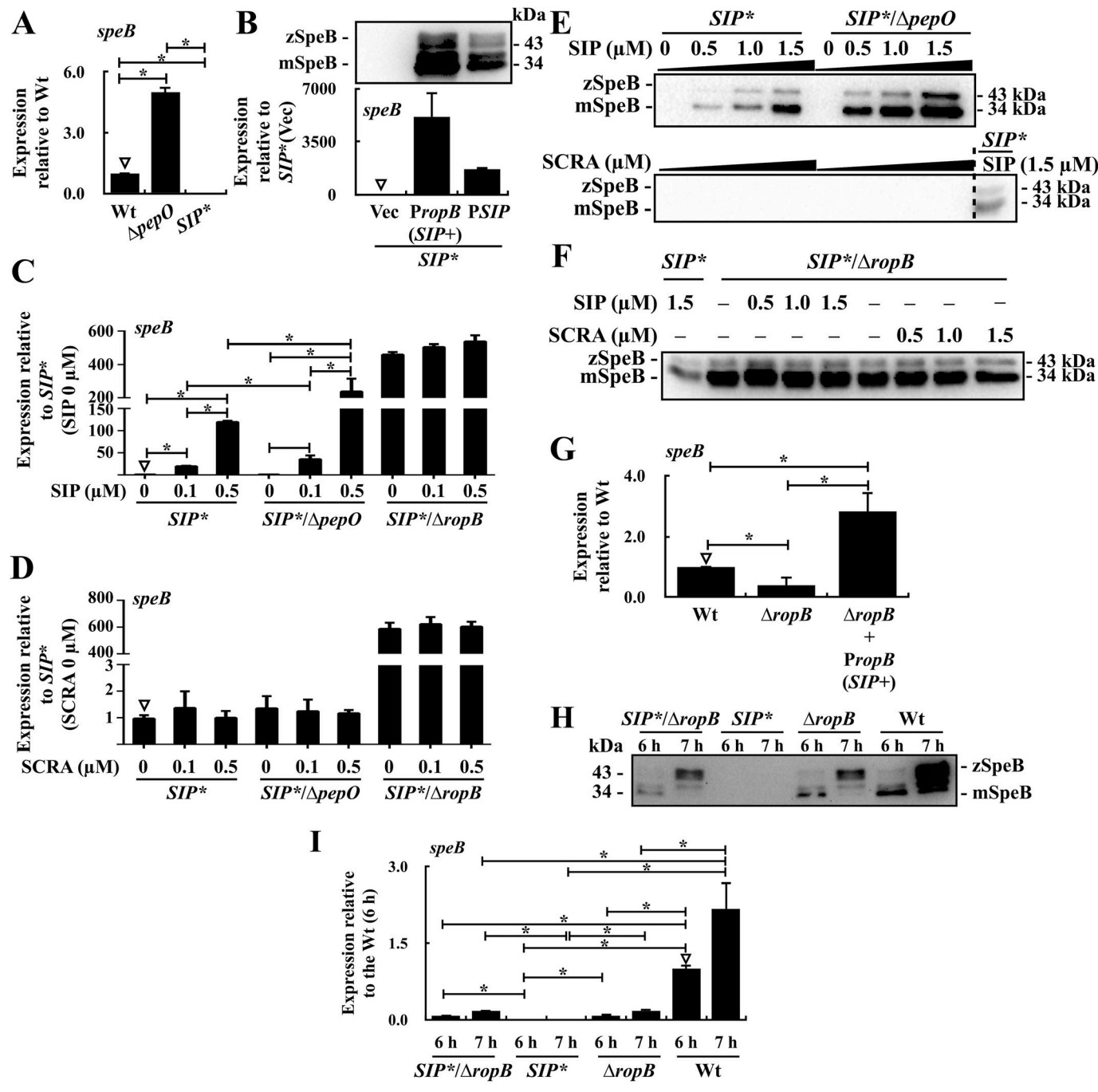

**Figure 1. The expression of *speB* in the wild-type strain (Wt), ∆*ropB* mutant, *SIP\** mutant, ∆*pepO* mutant, *SIP*/∆*pepO* mutant, *SIP\**/∆*ropB* mutant, and the *SIP* and *ropB* *trans*-complementary strains treated with different concentrations of the synthetic SIP and scramble peptide (SCRA).**
**(A)** The transcription of *speB* in the wild-type strain and its *pepO* (∆*pepO*) and SIP-inactivated (*SIP\**) mutants. **(B)** The expression of *speB* in the *SIP* mutant [with the empty vector (Vec)] and its *SIP* (P*SIP*) and *ropB* with its native promoter [P*ropB* (*SIP*+)] *trans*-complementary strains. **(C, D, E)** The transcription of *speB* and the expression of SpeB in the *SIP\** mutant, *SIP\**/∆*pepO* mutant, and *SIP\**/∆*ropB* mutant under SIP and SCRA treatments. **(F)** The expression of SpeB in the *SIP\** mutant and the *SIP\**/∆*ropB* mutant in the treatment of different concentrations of SIP and SCRA. **(G)** The transcription of *speB* in the wild-type strain, the *ropB* isogenic mutant (∆*ropB*), and the *ropB* *trans*-complementary strain [P*ropB* (*SIP*+)]. **(H, I)** The expression of SpeB and the transcription of *speB* in the wild-type strain, ∆*ropB* mutant, *SIP\** mutant, and *SIP\**/∆*ropB* mutant after 6–7 h of incubation. Culture supernatant was used for Western blot analysis. zSpeB, zymogen form of SpeB; mSpeB, mature form of SpeB. Bacterial RNA was extracted for real-time quantitative PCR (RT–qPCR) analysis. The expression of *speB* was normalized to that of *gyrA*. *P < 0.05. Source data are available for this figure.

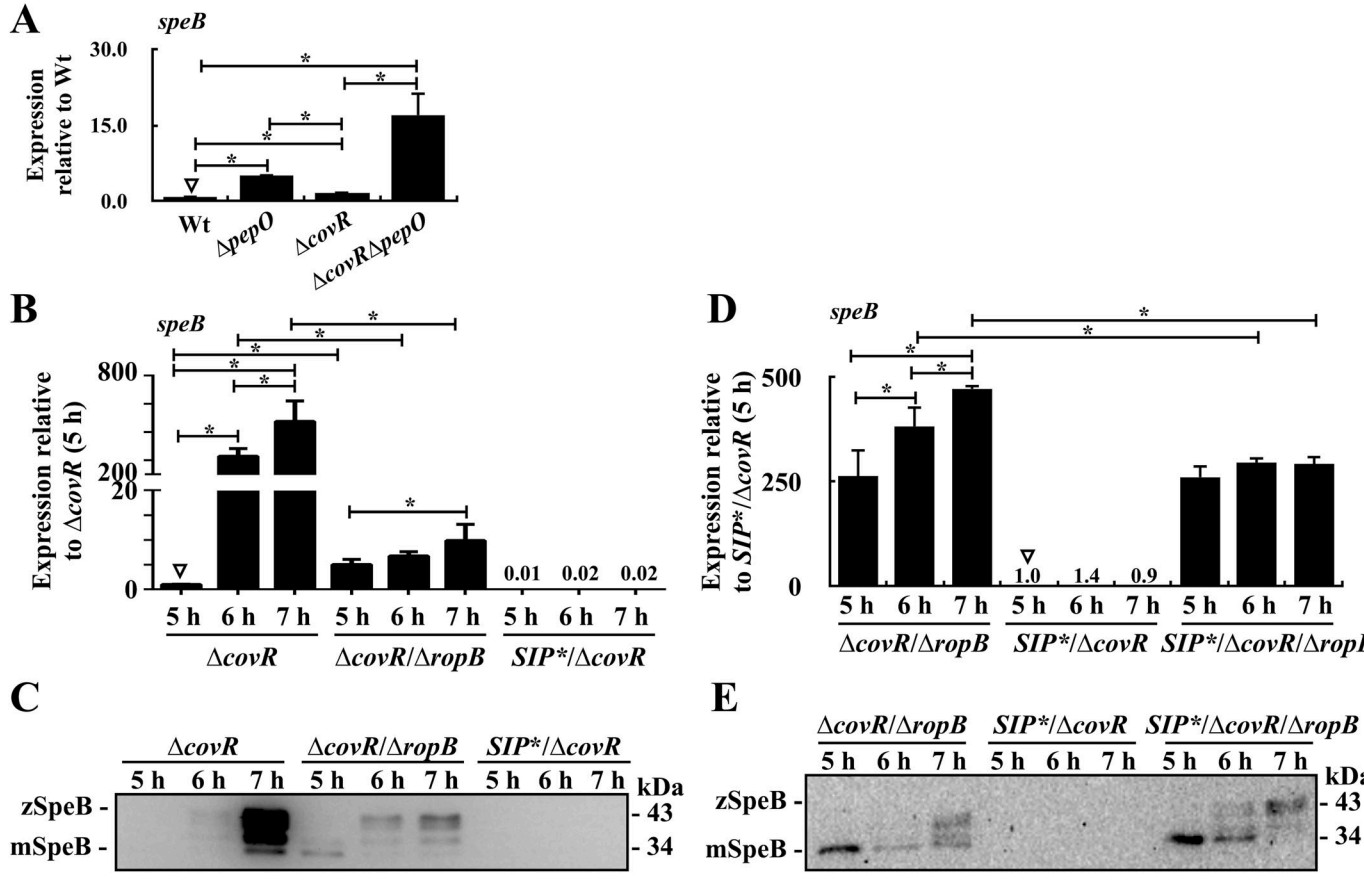

**Figure 2.  Expression of SpeB in the wild-type strain, *pepO* mutant (Δ*pepO*), *covR* mutant (Δ*covR*), Δ*covR*/Δ*pepO* mutant, *SIP\**/Δ*covR* mutant, Δ*covR*/Δ*ropB* mutant, and *SIP\**/Δ*covR*/Δ*ropB* mutant.**
**(A)** The expression of *speB* in the wild-type strain and its *pepO* mutant, *covR* mutant, and Δ*covR*/Δ*pepO* mutant. **(B, C)** Transcription of *speB* and the expression of SpeB in the *covR* mutant, Δ*covR*/Δ*ropB* mutant, and *SIP\**/Δ*covR* mutant. **(D, E)** Transcription of *speB* and expression of SpeB in the Δ*covR*/Δ*ropB* mutant, *SIP\**/Δ*covR* mutant, and *SIP\**/Δ*covR*/Δ*ropB* mutant. Culture supernatant was used for Western blot analysis. zSpeB, zymogen form of SpeB; mSpeB, mature form of SpeB. Bacterial RNA was extracted for real-time quantitative PCR (qPCR) analysis. The expression of *speB* was normalized to that of *gyrA*. *P < 0.05.
Source data are available for this figure.

mutant (Fig 2D and E), suggesting that, in the absence of SIP, RopB inhibits *speB* transcription in the *covR* mutant.

### RopB represses SpeB expression in the CovS kinase-inactivated mutant

Unlike the *covR* mutant, the CovS-inactivated [the *covS*-deletion (Δ*covS*) and the kinase-inactivated (CovS$_{H280A}$)] mutants still produce the non-phosphorylated CovR protein (Fig 3A) that represses *ropB* transcription (Chiang-Ni et al, 2019a; Finn et al, 2021; Horstmann et al, 2022). Therefore, *speB* is derepressed in the *covR* mutant but repressed in the *covS* mutant (Sumby et al, 2006; Chiang-Ni et al, 2016, 2019a; Finn et al, 2021; Horstmann et al, 2022). Similar to the *covR* mutant, Western blot analysis showed that PepO expression was higher in the *covS* mutant than in the wild-type strain (Fig 3B). We also found that the *pepO*-deleted *covS* mutant (Δ*covS*/Δ*pepO*) expressed a higher level of *speB* than the *covS* mutant under the same concentration of SIP treatments (Fig 3C). These results indicate that PepO is involved in abrogating the SIP-

induced *speB* expression. Therefore, the up-regulation of PepO may have contributed to the low SIP concentration in the *covS* mutant.

Furthermore, the role of RopB in regulating *speB* expression was analyzed in the CovS kinase-inactivated mutant (CovS$_{H280A}$) and a CovS phosphatase-inactivated mutant (CovS$_{T284A}$). Consistent with our previous study (Chiang-Ni et al, 2019a), CovR phosphorylation was inactivated in the CovS$_{H280A}$ mutant but slightly increased in the CovS$_{T284A}$ mutant (Fig 3A). In addition, the transcription of *ropB* was repressed in the CovS$_{H280A}$ mutant compared with the wild-type strain and the CovS$_{T284A}$ mutant (Fig 3D). Next, the expression of SpeB in the wild-type strain, CovS$_{H280A}$ mutant, CovS$_{T284A}$ mutant, and their *ropB* mutants (CovS$_{H280A}$/Δ*ropB* and CovS$_{T284A}$/Δ*ropB*) were evaluated via Western blotting. As expected, SpeB expression in the *ropB* isogenic mutant (Δ*ropB*) and CovS$_{T284A}$/Δ*ropB* mutant were down-regulated compared with that in their parental strains (Fig 3E). The CovS$_{H280A}$ mutant showed low levels of *ropB* transcription (Fig 3D); however, SpeB expression was completely repressed (Fig 3E). Notably, the expression of SpeB in the CovS$_{H280A}$/Δ*ropB* mutant was increased to a level similar to that in the *ropB* mutant and CovS$_{T284A}$/Δ*ropB* mutant (Fig 3E). At the transcriptional

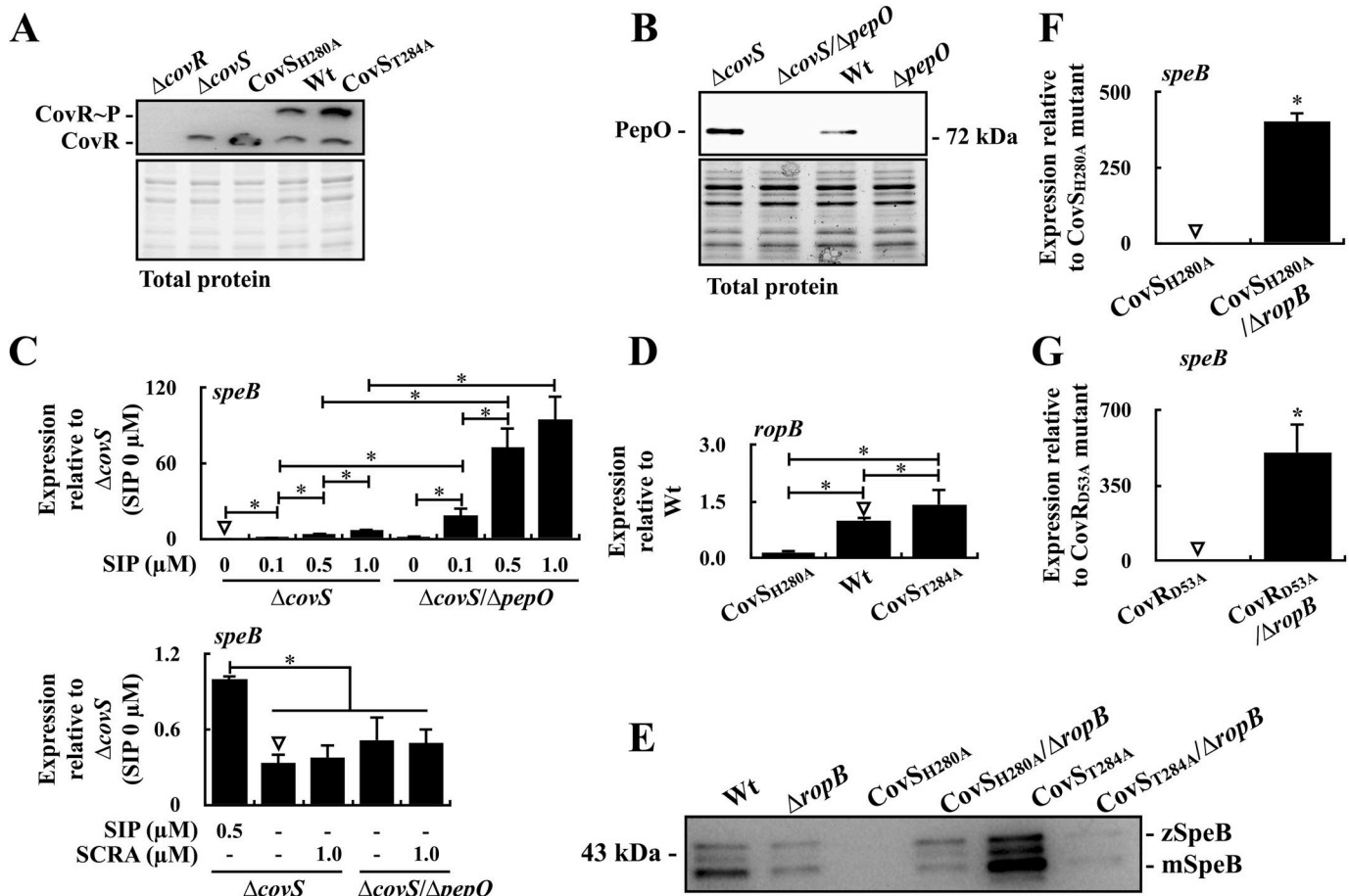

**Figure 3. The expression of PepO in the *covS* mutant (Δ*covS*) and the expressions of *speB* and *ropB* in the *covS* mutants, the CovR D53A substitution mutant (CovR$_{D53A}$), their *ropB* mutants, and the Δ*covS*/Δ*pepO* mutant.**
**(A)** The phosphorylation level of CovR in the wild-type strain (Wt), CovS kinase-inactivated (CovS$_{H280A}$) mutant, and CovS phosphatase-inactivated (CovS$_{T284A}$) mutant. The Δ*covS* mutant (the mutant that cannot phosphorylate CovR) and the *covR* mutant (Δ*covR*) were used as experimental controls. The total protein is used as the internal loading control. **(B)** The expression of PepO in the wild-type strain, its *pepO* mutant (Δ*pepO*), Δ*covS* mutant, and the Δ*covS*/Δ*pepO* mutant. The lower panel shows the total protein as the internal loading control. **(C)** The transcription of *speB* in the Δ*covS* mutant and Δ*covS*/Δ*pepO* mutant under the synthetic SIP and scramble peptide (SCRA) treatments. **(D)** The transcription of *ropB* in the wild-type strain, CovS$_{H280A}$ mutant, and CovS$_{T284A}$ mutant. **(E)** The expression of SpeB in the wild-type strain, CovS$_{H280A}$ mutant, CovS$_{T284A}$ mutant, and their *ropB* mutants. **(F)** The expression of *speB* in the CovS$_{H280A}$ mutant and its *ropB* mutant (CovS$_{H280A}$/Δ*ropB*). **(G)** The transcription of *speB* in the CovR D53A substitution mutant (CovR$_{D53A}$) and its *ropB* mutant (CovR$_{D53A}$/Δ*ropB*). Culture supernatant was used for Western blot analysis. zSpeB, zymogen form of SpeB; mSpeB, mature form of SpeB. Bacterial RNA was extracted for real-time quantitative PCR (qPCR) analysis. The expression of *ropB* and *speB* was normalized to that of *gyrA*. *$P < 0.05$.
Source data are available for this figure.

level, the expression of *speB* was significantly up-regulated in the CovS$_{H280A}$/Δ*ropB* mutant compared with that in the CovS$_{H280A}$ mutant (Fig 3F). These results indicate that SpeB expression in the CovS$_{H280A}$ mutant was inhibited by RopB.

CovS phosphorylates the D53 residue of CovR (Dalton & Scott, 2004). Similar to the *covS* mutant, *speB* expression is repressed in the CovR D53A substituted (CovR$_{D53A}$) mutant (Chiang-Ni et al, 2019a). To demonstrate the role of RopB in regulating *speB* expression in the CovR non-phosphorylated mutant, the expression of *speB* in the CovR$_{D53A}$ mutant and its *ropB* isogenic mutant (CovR$_{D53A}$/Δ*ropB*) was compared. The results showed that the expression of *speB* in the CovR$_{D53A}$/Δ*ropB* mutant was derepressed compared with that in the CovR$_{D53A}$ mutant (Fig 3G), suggesting that the transcription of *speB* was inhibited by RopB in the CovR$_{D53A}$ mutant.

## Apo-RopB represses the expression of *speB* and its co-transcripts in the GAS transcriptome

To elucidate the role of apo-RopB in the GAS transcriptome, RNA was extracted from the wild-type A20 strain, its *SIP** mutant, and the *SIP**/Δ*ropB* mutant and analyzed by RNA sequencing. In comparison with the wild-type strain, only three genes, *speB*, *spi*, and *M5005_Spy1733*, were significantly (*q* value < 0.05) down-regulated in the *SIP** mutant (closed points in Fig S1 and Table S1). Furthermore, in the *SIP**/Δ*ropB* mutant, the expression of *speB* and *spi* was significantly down-regulated (*q* value < 0.05) compared with the wild-type strain (Fig 4A and Table S2) but up-regulated when compared with that in the *SIP** mutant (Fig 4B and Table S3), indicating that apo-RopB and RopB-SIP would act differently on

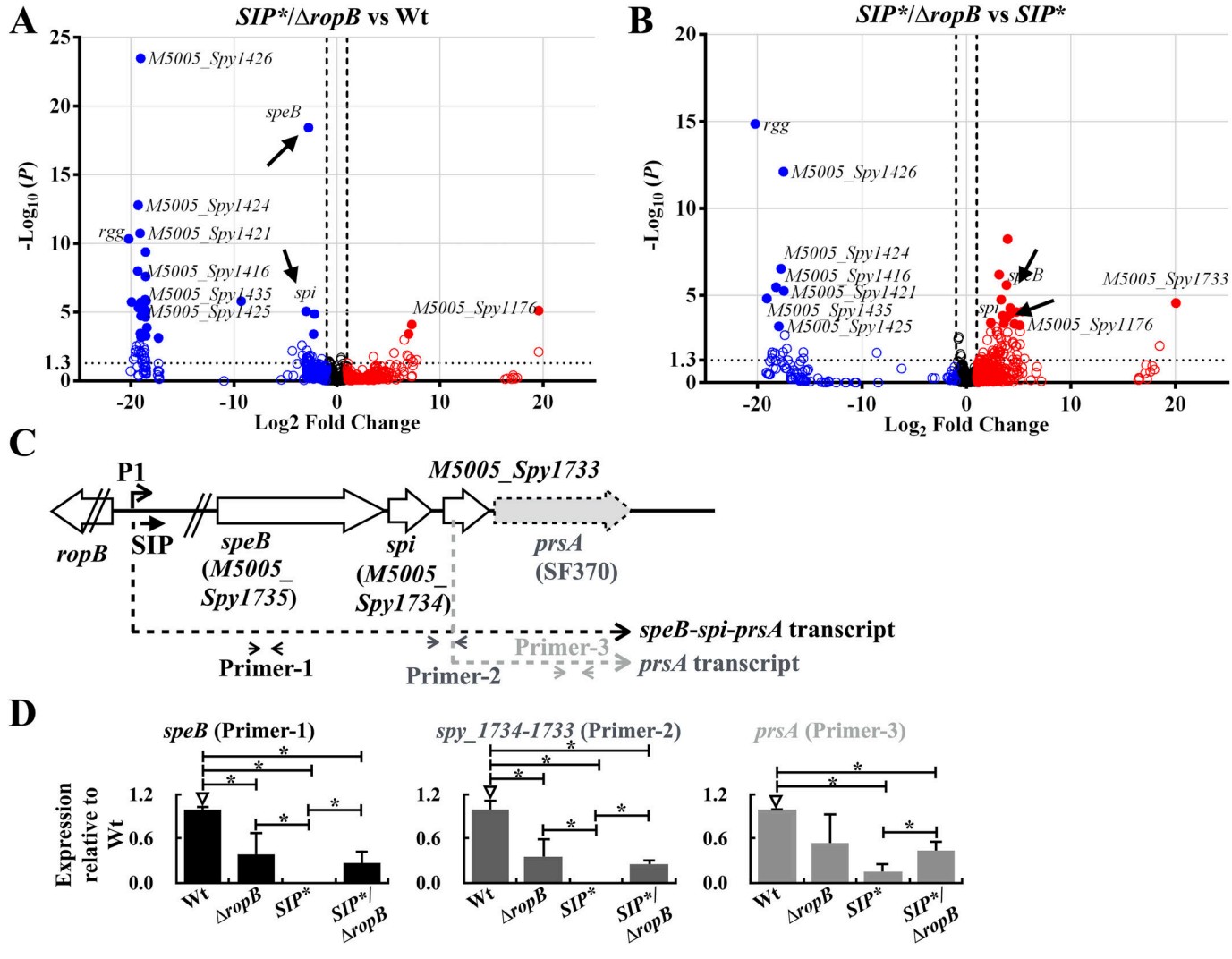

**Figure 4. RNA-sequencing analysis of the wild-type strain (Wt), *SIP\** mutant, and *SIP\**/Δ*ropB* mutant, and the expression of *speB* and its co-transcripts in these strains.**
**(A, B)** The genes those were differentially expressed in (A) *SIP\**/Δ*ropB* mutant versus the wild-type strain (Wt) and (B) *SIP\**/Δ*ropB* mutant versus the *SIP\** mutant are visualized by the volcano plot. Blue circles and red circles indicate the down-regulated and up-regulated genes, respectively, in the (A) *SIP\**/Δ*ropB* mutant compared with that of the wild-type strain and (B) *SIP\**/Δ*ropB* mutant compared with that of the *SIP\** mutant (*P* < 0.05). The solid circles indicate that the expression difference is statistically significant (adjusted *P*-value, *q* value < 0.05). **(C)** Schematic representation of the *speB*, *spi*, and *prsA* genes (arrows). The *speB* and its co-transcripts (dashed lines) and the location of primers (Primer-1–Primer-3) used for detecting *speB* and its co-transcripts are indicated. The genes and their annotations are indicated according to MGAS5005 (NCBI Accession: CP000017.2; the open arrows) and SF370 (NCBI Accession: NC_002737.2; the gray arrow). **(D)** The expression of *speB* and its co-transcripts in the wild-type strain, the *ropB* mutant, *SIP\** mutant, and *SIP\**/Δ*ropB* mutants detected by Primer-1–Primer-3. Bacterial RNA was extracted for sequencing and real-time quantitative PCR (qPCR) analyses. The expression of the target transcript was normalized to that of *gyrA*. *\*P* < 0.05.
Source data are available for this figure.

regulating *speB* and *spi* expression. *spi* and *M5005_Spy1733* are downstream of *speB*, and *M5005_Spy1733* has been annotated as a hypothetical protein in the *emm*1-type MGAS5005 strain (open arrow, Fig 4C; NCBI Accession: CP000017.2). However, in the *emm*1-type SF370 strain, the *prsA* gene was annotated instead of *M5005_Spy1733* (gray arrow; Fig 4C; NCBI Accession: NC_002737.2). Furthermore, Ma et al (2006) showed that the *prsA* gene is transcribed by its promoter (1.2 kb) or co-transcribed with *speB* and *spi* (*speB-spi-prsA*, 3.2–3.8 kb; Fig 4C) by the *speB* promoter. We used primers targeting *speB*, the intergenic regions of *M5005_Spy1734* (*spi*), *M5005_Spy1733*, and *prsA* to verify whether apo-RopB

represses *speB-spi-prsA* and *prsA* transcription. RT–qPCR analysis showed that the transcription of *speB* and *speB-spi-prsA* was down-regulated in the *ropB* mutant compared with that in the wild-type strain (Fig 4D). Further, in support of the RNA-Seq results, the expression of these genes was up-regulated in the *SIP\**/Δ*ropB* mutant compared with that in the *SIP\** mutant (Fig 4D). Noticeably, the expression of *speB* and *speB-spi-prsA* in the *SIP\**/Δ*ropB* mutant was up-regulated by ~415-fold and 57-fold, respectively, compared with the *SIP\** mutant. Although *prsA* is co-transcribed with *speB* and *spi* (Ma et al, 2006), the expression of *prsA* was increased by only ~threefold in the *SIP\**/Δ*ropB* mutant compared with that in the *SIP\**

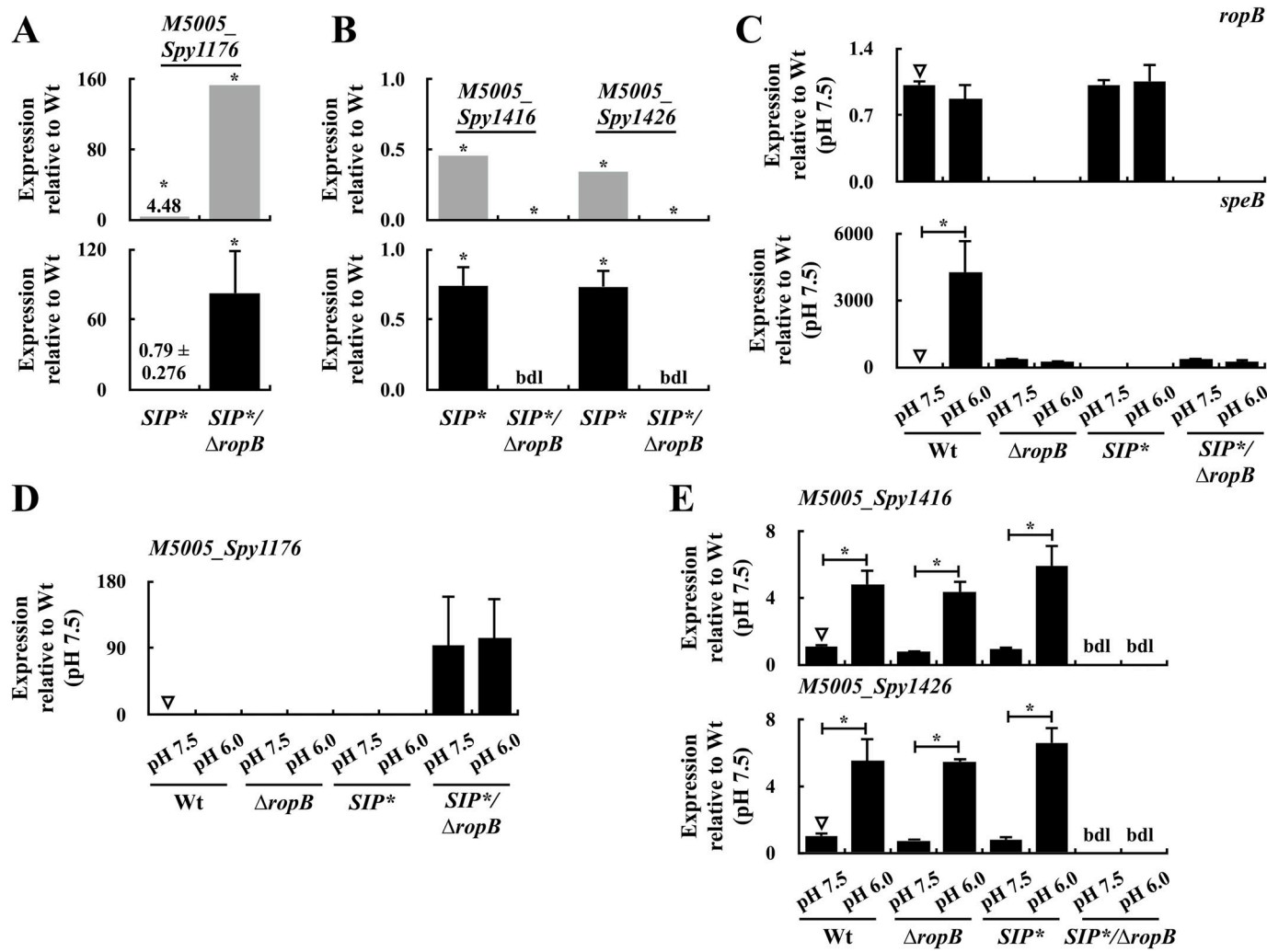

**Figure 5. Expression of RopB-SIP-regulated genes in the wild-type A20 strain, its *ropB* mutant (Δ*rpoB*), *SIP\** mutant, and *SIP\**/Δ*ropB* mutant in the early stationary phase and under the neutral and acidic conditions.**
**(A, B)** Expression of (A) *M5005_Spy1176* (negatively regulated by RopB-SIP) and (B) *M5005_1416* and *M5005_Spy1426* (positively regulated by RopB-SIP) in A20, the *SIP\** mutant, and *SIP\**/Δ*ropB* mutant in the early stationary phase of growth (O.D.$_{600}$ = 1.0). The upper and lower panels of (A, B) show the results from RNA-seq analysis and real-time quantitative PCR (qPCR) analysis, respectively. **(C, D, E)** Expression of (C) *ropB* and *speB*, (D) *M5005_Spy1176*, and (E) *M5005_1416* and *M5005_Spy1426* in A20, the *SIP\** mutant, and *SIP\**/Δ*ropB* mutant under neutral (pH 7.5) and acidic (pH 6.0) conditions. RNAs were extracted for qPCR analysis. The expression of target genes was normalized to that of *gyrA*. bdl, below detection limit. *$P < 0.05$.
Source data are available for this figure.

mutant (Fig 4D). These results suggested that apo-RopB plays a minor role in regulating *prsA* expression and represses the expression of only *speB* and its co-transcripts in the GAS transcriptome.

### SIP-mediated quorum-sensing regulation acts predominantly on the *speB* operon

The results of the transcriptomic analysis suggest that SIP could be a signal that explicitly controls the expression of *speB* and its co-transcripts. To test this, the role of SIP in regulating RopB-regulated genes was analyzed. The expression of *M5005_Spy1176* and six phage-related gene mutants was down-regulated and up-regulated in the wild-type A20 strain, respectively, compared

with that in the *SIP\**/Δ*ropB* mutant (Fig 4A and B and Tables S2 and S3). In line with the RNA-Seq results (Fig 5A, the upper panel), the RT–qPCR analysis showed that the expression of *M5005_Spy1176* was down-regulated in the wild-type and the *SIP\** mutant strains compared with that in the *SIP\**/Δ*ropB* mutant (Fig 5A, the lower panel), and the expression of *M5005_Spy1416* and *M5005_Spy1426* was undetectable in the *SIP\**/Δ*ropB* mutant (Fig 5B, the lower panel), indicating that the expression of these genes was regulated by RopB. Noticeably, the inactivation of SIP had a minor impact on the expression of these genes (the fold change in expression was less than twofold, Fig 5A and B, lower panels).

RopB binds to SIP under acidic pH conditions (Do et al, 2019). To evaluate the role of SIP in regulating RopB-regulated genes, the expressions of *ropB*, *speB*, *M5005_Spy1176*, *M5005_Spy1416*, and

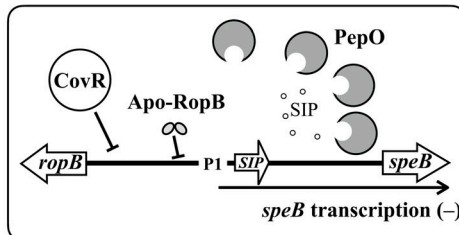

**Figure 6. Hypothetical models of *speB* regulation in the *covR* and *covS* mutants.** The expression of *ropB* and *pepO* are repressed by CovR. Although the up-regulated PepO would degrade SIP in the *covR* mutant, the effect of PepO degradation could be compensated by the derepression of *ropB* and *SIP*, and the SIP-bound RopB (RopB-SIP) could activate *speB* transcription. In the *covS* mutant, the expression of *pepO* is up-regulated, whereas that of *ropB* is repressed by the non-phosphorylated CovR. Therefore, the repression of *speB* in the *covS* could be mediated by the PepO-dependent SIP degradation and the SIP-free RopB (apo-RopB)-dependent transcriptional repression.

*M5005_Spy1426* in the wild-type strain, *SIP** mutant, and their *ropB* mutants were checked under neutral (pH 7.5) and acidic (pH 6.0) conditions. The expression levels of *ropB* in the wild-type A20 strain and the *SIP** mutant were similar under neutral and acidic conditions (Fig 5C), and this acted as an experimental control. The *speB* expression was only induced in the wild-type strain but not in the *ropB* and *SIP** mutants (Fig 5C), indicating that *speB* expression under acidic conditions is activated upon the binding of RopB to SIP. Upon comparing the wild-type A20 strain, the *ropB* mutant, and the *SIP** mutant, the expression of *M5005_Spy1176* was found to be up-regulated, whereas that of *M5005_Spy1416/Spy1426* was down-regulated in the *SIP*/ΔropB* mutant (Fig 5D and E). These results suggest that RopB-SIP has a crucial role in regulating the expression of these genes. However, SIP was not involved in regulating the expression of these genes under neutral and acidic conditions (Fig 5D and E). We also examined the expression of other RopB-SIP-regulated genes, including *M5005_Spy1189*, *adh2*, and *M5005_Spy0023*, by RT–qPCR. We found that SIP did not play a role in regulating the expression of these genes under neutral and acidic conditions (Fig S2).

## Discussion

RopB is a quorum-sensing protein that binds to SIP under acidic conditions to activate *speB* transcription (Do et al, 2017, 2019). Finn et al (Finn et al, 2021) suggested that CovR might regulate *speB* expression indirectly through RopB. Our previous study showed that SIP could be degraded by the CovR/CovS-controlled endopeptidase PepO (Shi et al, 2022). Therefore, increased PepO expression in *covR* and *covS* mutants could down-regulate SIP-induced SpeB expression. Transcription of *ropB* and *speB* is repressed by both phosphorylated and non-phosphorylated CovR (Miller et al, 2001; Chiang-Ni et al, 2019a). In the *covR* mutant, the effect of PepO-mediated SIP degradation was compensated by the derepression of *ropB* and *speB*, resulting in the up-regulation of *speB* in the stationary phase of growth (Figs 6 and S3) (Shi et al, 2022). In the *covS* mutant, the transcription of *ropB* is repressed by non-phosphorylated CovR (Chiang-Ni et al, 2019a; Finn et al, 2021). In this study, we further demonstrated that RopB functions as a transcriptional repressor of *speB* in the absence of SIP. Therefore,

the repression of *speB* transcription in the *covS* mutant is not only mediated by the down-regulation of *ropB* transcription but also by RopB-dependent transcriptional repression (Fig 6).

RopB binds to the *speB* promoter, and this interaction has been considered essential for activating *speB* transcription (Lyon et al, 1998; Neely et al, 2003; Makthal et al, 2016). Do et al (2017) proposed that in the log phase of growth, the inhibition peptide Vfr binds to RopB (Shelburne et al, 2011) to inhibit the RopB–DNA interaction and abolishes *speB* transcription. Our study showed that the deletion of *ropB* in the *SIP** mutant derepressed the transcription of *speB*, indicating that the interaction between RopB and the *speB* promoter is not essential for *speB* transcription. We also show that disrupting the interaction between RopB and SIP or decreasing the intracellular concentration of SIP mediates an apo-RopB-dependent down-regulation of *speB* transcription. The *speB* transcription is increased dramatically under acidic stimuli or in the stationary phase of growth compared with that in the neutral pH or log phase of growth (Loughman & Caparon, 2006; Chiang-Ni et al, 2012). In addition, SpeB is the most abundant protein secreted in the GAS culture supernatants. The results of this study suggest that RopB would not only augment the *speB* expression under acidic culture and stationary phase growth conditions but also play a critical role in preserving energy by preventing the transcriptional leakage of *speB* under neutral pH and log phase growth conditions in an SIP-dependent manner.

RopB engages in high-affinity interactions with SIP under acidic conditions, suggesting that the pH- and growth-phase dependence of *speB* expression is because of the influence of pH on the association between RopB and SIP (Unnikrishnan et al, 1999; Loughman & Caparon, 2006; Chiang-Ni et al, 2012; Do et al, 2017, 2019). The RNA-seq and RT–qPCR analyses in this study show that under regular and acidic culture conditions, SIP-mediated regulation acts predominantly on the transcription of *speB* and its co-transcripts. Do et al (2017) showed that in the *SIP**-inactivated mutant, the expression of *speB* (SpyM3_1742) and its downstream proteins *spi* (SpyM3_1741) and *M3_1743* (SpyM3_1743) were down-regulated by over 1,000fold compared with that in the wild-type MGAS10870 strain. The fold change of other identified genes in their RNA-seq analysis was between 4.4 and 2.0 (Do et al, 2017), supporting that the SIP signal would have the most significant impact on controlling the *speB* transcription. However, the SIP-regulated

genes in MGAS10870 (*emm*3) and A20 (*emm*1) were not identical. Lynskey et al (2015) demonstrated that a premature stop codon in the *rocA* gene was found in the M3 serotype strains, including MGAS10870 (Jain et al, 2017). RocA is an accessory protein that inhibits the phosphatase activity of CovS (Chiang-Ni et al, 2020). Furthermore, the study also showed that the M1 serotype GAS strains had high levels of phosphorylated CovR compared with that of the M3 serotype strains (Horstmann et al, 2015). CovR/CovS can modulate the regulatory activity of RopB by controlling *pepO* transcription. Therefore, the inconsistent RNA-sequencing results from the M1 and M3 type strains could be related to different levels of phosphorylated CovR. These results also reveal complicated interactions between the two-component CovR/CovS system and the RopB-SIP quorum-sensing system in the GAS regulatory network.

The expression of *speB*, *spd3*, and *grab* were repressed in the *covS* mutant compared with that in the wild-type strain, suggesting that CovS phosphorylates CovR to activate the expression of these genes (Trevino et al, 2009; Tran-Winkler et al, 2011). Horstmann et al (2022) suggested that in contrast to the transcriptional repression of phosphorylated CovR, predominantly mediated by a direct mechanism, phosphorylated CovR-mediated transcriptional activation is indirect and could be complex. This study further showed that the repression of *speB* in the *covS* mutant was mediated by apo-RopB, indicating that non-phosphorylated CovR-mediated *speB* repression is a consequence of the interaction between the CovR/CovS and RopB-SIP systems. Nonetheless, the repression of *spd3* and *grab* in the *covS* mutant was mediated by a RopB-independent mechanism (data not shown), suggesting that multiple regulatory pathways are involved in non-phosphorylated CovR-mediated transcriptional regulation (Finn et al, 2021).

This study showed that RopB functions as a transcriptional repressor of *speB* in the absence of SIP, revealing unidentified roles of RopB in regulating *speB* expression. Do et al (2017) showed that purified apo-RopB forms a homodimer and can bind the *speB* promoter with activity similar to that of RopB-SIP in vitro. Therefore, we suggest that the RopB dimer could form different structures with the *speB* promoter in the presence or absence of SIP in vivo, and these RopB-DNA structures are crucial for modulating *speB* transcription. Unfortunately, this hypothesis cannot be further verified because modification of the *ropB*-*speB* intergenic region abolishes *speB* transcription (Fig S4). The underlying mechanisms by which apo-RopB and RopB-SIP act differently to control *speB* expression remain to be investigated.

# Materials and Methods

## Bacterial strains and culture conditions

GAS A20 (*emm*1-type) bacteria were isolated and cultured as described previously (Chiang-Ni et al, 2009). Strain AP3 is an animal passage isolate of A20 with a frameshift deletion in the *covS* gene (Chiang-Ni et al, 2016). GAS strains were cultured on trypticase soy agar containing 5% sheep blood or in tryptic soy broth (Becton Dickinson and Company) supplemented with 0.5% yeast extract (TSBY). *Escherichia coli* DH5α was purchased from Yeastern (Yeastern Biotech Co., Ltd.) and was cultured in lysogeny broth (LB) at 37°C with vigorous aeration. SpeB-inducing peptide (SIP; MWLLLLFL; purity: 94.469%) and scrambled control peptide (SCRA, LLFLWLLM; purity: 92.822%) (Do et al, 2017) were purchased from Leadgene Biomedical Inc. These synthetic peptides were suspended in 100% DMSO to prepare a 10 mM stock solution and stored at −20°C until use. Working solutions were prepared by diluting the stock solution with 25% DMSO. SIP- and SCRA-supplemented culture conditions have been described previously (Shi et al, 2022). Briefly, GAS strains were grown to $O.D._{600}$ = 0.8 in TSBY broth. Bacterial pellets were collected and incubated in an acidic TSBY broth (pH 6.0) supplemented with different concentrations of SIP and SCRA for 1 h. To treat bacteria with neutral and acidic broth, bacterial pellets were collected ($O.D._{600}$ = 0.4), resuspended in either pH 7.5 or 6.0 broths, and cultured for another 4 h. The bacterial strains and plasmids used in this study are listed in Table 1. When appropriate, the antibiotics chloramphenicol (25 μg/ml for *E. coli* and 3 μg/ml for GAS) and spectinomycin (100 μg/ml) were used for selection.

## DNA and RNA manipulations

Bacterial genomic DNA and RNA extractions and reverse transcription were performed as previously described (Wang et al, 2013). Real-time PCR was performed in a 20 μl reaction mixture containing 1 μl of cDNA, 0.8 μl of primers (10 μM), and 10 μl of SensiFAST SYBR Lo-ROX pre-mixture (Bioline Ltd.) according to the instructions of the manufacturer. Biological replicates were performed using two to three independent RNA preparations in duplicate. The expression level of each target gene was normalized to *gyrA* and analyzed using the ΔΔCt method (QuantStudio 3 System; Thermo Fisher Scientific Inc.). All values of the control and experimental groups were divided by the mean of the control samples before statistical analysis (Valcu & Valcu, 2011). Primers used for real-time PCR analysis (Table S4) were designed using Primer3 (v.0.4.0, http://frodo.wi.mit.edu) based on the MGAS5005 sequence (NCBI accession number: CP000017.2). RNA samples were analyzed by RNA-sequencing (Welgene Biotech). SureSelect XT HS2 mRNA library preparation kit (Agilent) was used for library construction, followed by size selection using AMPure XP beads (Beckman Coulter). The sequence was determined using Illumina sequencing-by-synthesis technology (Illumina). Sequencing data (FASTQ reads) were generated using Welgene Biotech's pipeline based on the Illumina base-calling program bcl2fastq v2.20. The adjusted *P*-value (*q*-value) cut off to 0.05 (DESeq with non-grouped sample using blind mode) was set for discovering differentially expressed genes.

## Construction of the *ropB*-deletion, *pepO*-deletion, and SIP-inactivation mutants

To construct the *ropB* mutant, the *ropB* gene with its upstream (485 bp) and downstream (490 bp) regions was amplified using the primers ropB-F-5 and ropB-R-4 (Table S4). The PCR amplicon was digested with restriction enzyme (*Sph*I) and ligated into the temperature-sensitive vector pCN143 (Chiang-Ni et al, 2016). The *ropB* gene was removed via inverted PCR using the primers ropB-

**Table 1. Plasmids and strains used in this study.**

| Plasmid or strain | Description[a] | Reference or source |
|---|---|---|
| **Plasmids** | | |
| pDL278 | *E. coli* – *Streptococcus* shuttle vector | Chiang-Ni et al (2012) |
| pCN138 | pDL278::*ropB* (with its native promoter) | Chiang-Ni et al (2016) |
| pCN143 | Temperature-sensitive vector | Chiang-Ni et al (2016) |
| pCN146 | pCN143::*ropB*Δ*cat* | This study |
| pCN161 | pCN143::CovS$_{T284A}$ | Chiang-Ni et al (2019b) |
| pCN210 | pCN143::*pepO*Δ*cat* | Shi et al (2022) |
| pCN215 | pCN143::*SIP** | Shi et al (2022) |
| pCN228 | pDL278::p*speB* | This study |
| pCN230 | pCN143::p*speB* (P_del-1) | This study |
| pCN231 | pCN143::p*speB* (P_del-2) | This study |
| pCN232 | pCN143::p*speB* (P_del-3) | This study |
| pCN235 | pCN143::p*speB* (P2_del) | This study |
| **Strains** | | |
| A20 | *emm*1-type wild-type strain | Chiang-Ni et al (2009) |
| AP3 | A20 animal-passage, *covS* frameshift-deletion strain (Δ*covS*) | Chiang-Ni et al (2016) |
| SW656 | A20 Δ*covR* | Chiang-Ni et al (2016) |
| SCN128 | A20 CovR D53A substitution (CovR$_{D53A}$) mutant | Chiang-Ni et al (2016) |
| SCN142 | A20 Δ*ropB* | This study |
| SCN143 | A20 Δ*covR*/Δ*ropB* | This study |
| SCN152 | A20 CovS$_{H280A}$ mutant | Chiang-Ni et al (2017) |
| SCN167 | A20 CovS$_{T284A}$ mutant | Chiang-Ni et al (2019b) |
| SCN203 | AP3 Δ*ropB* | This study |
| SCN248 | A20 CovS$_{H280A}$/Δ*ropB* | This study |
| SCN249 | A20 CovS$_{T284A}$/Δ*ropB* | This study |
| SCN250 | A20 CovR$_{D53A}$/Δ*ropB* | This study |
| SCN274 | A20 *SIP**/Δ*covR*/Δ*pepO* | Shi et al (2022) |
| SCN281 | A20 Δ*pepO* | This study |
| SCN305 | A20 *SIP** | Shi et al (2022) |
| SCN306 | A20 *SIP**/Δ*covR* | Shi et al (2022) |
| SCN318 | AP3 Δ*pepO* | Shi et al (2022) |
| SCN312 | A20 *SIP**/Δ*ropB* | This study |
| SCN328 | A20 Δ*covR*/Δ*pepO* | Shi et al (2022) |
| SCN331 | A20 *SIP**/Δ*covR*/Δ*ropB* | This study |
| SCN339 | A20 *SIP**/Δ*pepO* | This study |
| SCN364 | A20 p*speB* (P_del-2) | This study |
| SCN366 | A20 p*speB* (P_del-1) | This study |
| SCN367 | A20 p*speB* (P_del-3) | This study |
| SCN372 | A20 p*speB* (P2_del) | This study |

[a]*cat*, chloramphenicol cassette; *SIP**: The translation start codon of SIP is mutated to TAG.

EcoRV-F and ropB-EcoRV-R (Table S4) and replaced with the chloramphenicol cassette from Vector 78 (Chiang-Ni et al, 2012) to generate pCN146 (Table 1). Plasmids used for constructing *pepO*-deletion mutants (pCN210) and SIP-inactivation mutants (pCN215) have been described previously (Shi et al, 2022). These plasmids were transformed into GAS strains via electroporation, and the transformants were selected as described previously (Chiang-Ni et al, 2016). Deletions of *ropB* and *pepO* and replacement of TAG in the *SIP* open reading frame were confirmed by Sanger sequencing.

### Construction of *SIP* and *ropB* trans-complementary strains

The *ropB* trans-complementary strain was constructed using a method described previously (Chiang-Ni et al, 2016). The open reading frame of SIP is located in the intergenic region between *ropB* and *speB* (Do et al, 2017). To construct the *SIP* trans-complementary strain, the intergenic region of *ropB* and *speB* was amplified using the primers PspeB-SacI-F-2 and PspeB-SacI-R-2 (Table S4), and the PCR product (956 bp) was ligated into pDL278 (Table 1). The constructed plasmid was designated pCN228 and transformed into *SIP\** mutants via electroporation.

### Western blot and Phos-tag Western blot

To detect phosphorylated CovR, bacteria were cultured in TSBY broth for 6 h, and then the bacterial cells were disrupted using a bead beater (Mini-Beadbeater; BioSpec Products Inc.). The bacterial cell lysate was centrifuged, and the supernatant was collected for analysis. Total protein (10 µg) was mixed with 6× protein loading dye, boiled for 5 min, and subjected to 12% SDS–PAGE. For Phos-tag Western blot analysis, the bacterial proteins were mixed with 6× protein loading dye (without boiling) and loaded into a 10% SDS–PAGE containing 10 µM of Phos-tag (Wako Pure Chemical Industries Ltd.) and 0.5 µM MnCl$_2$ (Chiang-Ni et al, 2016). To detect SpeB, the filtered (0.22 µm membrane filter; Millipore) culture supernatants were collected and subjected to 10% SDS–PAGE. Separated proteins were transferred onto polyvinylidene fluoride membranes (Millipore). The membranes were blocked with 5% skim milk in PBST buffer (PBS containing 0.2% vol/vol Tween-20) at 37°C for 1 h. CovR protein was detected using anti-CovR serum (Chiang-Ni et al, 2016), PepO was detected using a polyclonal anti-PepO antibody (Shi et al, 2022), and SpeB was detected using an anti-SpeB antibody (Toxin Technology, Inc.). After hybridization, the membrane was washed with PBST buffer and hybridized with a peroxidase-conjugated goat anti-rabbit IgG secondary antibody (Cell Signaling Technology, Inc.) at room temperature (25–28°C) for 1 h. The blots were developed using Pierce ECL Western blotting substrate (Thermo Fisher Scientific Inc.), and the signals were detected using a Gel Doc XR+ system (Bio-Rad Laboratories, Inc.).

### Statistical analysis

Statistical analyses were performed using Prism software version 5 (GraphPad Software, Inc.). Significant differences between multiple groups were determined using ANOVA. Post hoc tests for ANOVA were performed using Tukey's honest significance difference test. Statistical significance was set at $P < 0.05$. For RNA-sequencing analysis, the hypergeometric *P*-value was calculated as the probability of randomly drawing. The *P*-value was adjusted by false discovery rate for significance discovering (*q*-value). Differential gene expression with *P*-value and *q*-value < 0.05 was taken as significant.

## Supplementary Information

## Acknowledgements

We appreciate the assistance from Prof. Shuying Wang and Miss Yu-Tzu Chao (National Cheng Kung University, Taiwan) for EMSA analysis. This work was supported by a grant from the Chang Gung Memorial Hospital at Linkou, Taiwan (BMRPD19) and Ministry of Science and Technology, Taiwan (MOST 110-2628-B-182-012 and 111-2628-B-182-005).

### Author Contributions

C Chiang-Ni: conceptualization, resources, data curation, formal analysis, supervision, funding acquisition, project administration, and writing—original draft, review, and editing.
Y-W Chen: data curation, formal analysis, and investigation.
K-L Chen: data curation, formal analysis, and investigation.
J-X Jiang: data curation, formal analysis, and investigation.
Y-A Shi: data curation, formal analysis, and validation.
C-Y Hsu: data curation, formal analysis, and validation.
Y-YM Chen: conceptualization and supervision.
C-H Lai: conceptualization and supervision.
C-H Chiu: conceptualization, supervision, and funding acquisition.

### Conflict of Interest Statement

The authors declare that they have no conflict of interest.

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
