## [Reviewer comments · Life Science Alliance]

Life Science Alliance

RopB represses the transcription of *speB* in the absence of SIP in group A *Streptococcus*

Chuan Chiang-Ni, Yan-Wen Chen, Kai-Lin Chen, Jian-Xian Jiang, Yong-An Shi, Chih-Yun Hsu, Yi-Ywan M. Chen, Chih-Ho Lai, and Cheng-Hsun Chiu

DOI: <https://doi.org/10.26508/lsa.202201809>

Corresponding author(s): Chuan Chiang-Ni, Chang Gung University

Review Timeline:

Submission Date:	2022-11-08
Editorial Decision:	2022-12-19
Revision Received:	2023-02-07
Editorial Decision:	2023-03-06
Revision Received:	2023-03-17
Editorial Decision:	2023-03-20
Revision Received:	2023-03-23
Accepted:	2023-03-23

Scientific Editor: Novella Guidi

Transaction Report:

December 19, 2022

Re: Life Science Alliance manuscript #LSA-2022-01809-T

Prof. Chuan Chiang-Ni
Department of Microbiology and Immunology, College of Medicine, Chang Gung University, Taoyuan, Taiwan
Taiwan

Dear Dr. Chiang-Ni,

Thank you for submitting your manuscript entitled "RopB represses the transcription of speB in the absence of SIP in group A Streptococcus" to Life Science Alliance. The manuscript was assessed by expert reviewers, whose comments are appended to this letter. We invite you to submit a revised manuscript addressing the Reviewer comments.

Thank you for this interesting contribution to Life Science Alliance. We are looking forward to receiving your revised manuscript.

Sincerely,

B. MANUSCRIPT ORGANIZATION AND FORMATTING:

Reviewer #1 (Comments to the Authors (Required)):

Regulation of virulence gene expression in Group A Streptococcus (GAS) is under the control of the master regulator of virulence CovR. This regulation involves the direct and negative regulation of several key virulence genes, but also involves more complex mechanism at specific loci. This manuscript focus on the unusual positive regulation of the SpeB protease by CovR. This regulation involves a regulatory network with a quorum-sensing peptide (SIP), a CovR-regulated protease (PepO), and a SIP-binding activated transcriptional regulator (RopB). In this manuscript, the author show that a specific SIP inactivated mutant is not strictly similar to a RpoB deletion mutant for SpeB expression, suggesting that an apo-form of RpoB might become a repressor of speB transcription.

Major general comments:

It is unfortunate that this manuscript is difficult to read and therefore to evaluate. A recurrent issue is the use of strains, mutants, or genes without prior description. This is partly an editing problem, as some descriptions of mutants are provided in the text after, and not before, the interpretation of experiments. As a result, the overall impression is that of a highly specialized article that is not accessible to readers not directly involved in this specific regulation.

A second major editing issue is the absence of key directly relevant publications in the introduction, especially the Finn et al (mBio 2021, doi.org/10.1128/mBio.01642-21) showing that the expression of ropB and speB is repressed by nonphosphorylated CovR, and on the high variability of the ropB regulon in GAS strains. Despite being cited in the discussion, the Finn et al publication is essential in the introduction.

A third editing issue is the difficulties to interpret the experiments due to relative scales often used by the Authors. Usually the relative scales are against a mutant strain without providing first a comparative analysis between the WT and the mutant to infer the level of regulation.

It should be also noted that no genetic complementation of mutant phenotypes is provided in the manuscript. Especially, it is unclear if the SIP* mutant does not have secondary mutations in its genome, or if the introduced mutation does not impact the transcription of the operon, the stability of the RNA, or the translational efficiency of the other genes.

Comments:

Abstract: I can only suggest to re-write the abstract. The abstract is difficult to read without prior knowledge of the regulatory systems and their components and of the specific literature. For instance, genes are ill-defined. In addition, it is somehow unclear what have been previously published and what is the novelty.

Introduction line 75: I don't understand what "has not been inactivated but transformed" means.

Results, lines 82-96: please defined the strain used and the type of inactivating mutation for the SIP* mutant. For Fig. 1 A-D, please provided first the WT controls and included the single mutants (pepO, ropB, ...) . An option will be to first described the actual panel E. It will be also useful to describe for non-specialists what are the zymogen and mature forms of SpeB (zSpeB, and mSpeB). It is unclear what 'if the regulatory activity of RopB is inactivated in the absence of SIP' and 'would have an undefined role' means. Please rephrase them.

Results, lines 97-108 (corresponding to fig. 1): The results corresponding to Fig.1 are difficult to follow. The main results are: first a confirmation that PepO degrades SIP (natural and synthetic), and second that there is a difference of between ropB and sip* mutants for SpeB expression. The conclusion is that RopB could act as a transcriptional repressor of speB in the absence of SIP. I am not convince since other hypothesis can be made at this stage (indirect regulation, CovR regulation). The Authors should be more cautious in their hypotheses and experimental interpretation.

Results, lines 110-129 (corresponding to fig. 2): It will really help readers (and the reviewer) to have a wild-type control in fig. 2. It is also unclear how the overexpression of PepO in a Δ covR mutant has an effect in these conditions. It is unfortunate that pepO mutant where studies in fig. 1 and 3 but not in Fig. 2 where its effect is most likely the strongest.

Results, lines 131-157 (corresponding to fig. 3): Please define the covS mutant (line 133) before interpreting the result. Results obtained with different level of CovR phosphorylation are interesting. However, it is unfortunate that a CovR D53A mutant, a non

phosphorylable variant of CovR, not included in this study. This will be interesting, especially considering that the non-phosphorylated form of CovR appears to bind and regulate the speB and sip promoters (Finn et al 2021).

Results, lines 159-181 (corresponding to fig. 4): Once again, results are difficult to read and evaluate (double mutant versus WT and double mutant versus single mutant). It seems to me that the transcriptome, or at least qRT-PCR, of the single rpoB mutant is necessary to sustain the conclusion. Currently, it appears that the SIP* mutant specifically impacts the transcription of the sip-containing operon, raising concern on the stability of the RNA due to the mutation. In the absence of confirmatory experiments (maybe by exogenous complementation with synthetic or purified SIP), it is difficult to be convinced by the specific transcriptional regulation proposed by the Author.

Unfortunately, the direct binding of RpoB (Apo and with SIP) on the promoter has not been tested. This is a critical experiment that is necessary to not sustain the conclusion on transcriptional data only. The characterization of different binding sites for the Apo and the complexed RpoB will give a significant strength to the hypothesis of an activator becoming a repressor. The Authors should also consider (and tested) alternative hypothesis such as a competitive binding between CovR and RpoB.

Reviewer #2 (Comments to the Authors (Required)):

RopB represses the transcription of speB in the absence of SIP in group A streptococcus

In this article, the authors investigate the transcription/production of the key cysteine protease SpeB. They use a mutant strain of GAS in which the SpeB inducing peptide (SIP) has been inactivated. It has previously been shown that RopB-SIP interaction under acidic conditions is critical to speB expression. In figure 1 they show that inactivating RopB in the background of SIP inactivation increases speB expression and production. Figure 2 shows similar findings in a CovR background. Figure 3 introduces the endopeptidase pepO into speB regulation and shows differential impact of ropB inactivation dependent on CovS activity. The remainder of the manuscript focuses on the genome-wide impact of SIP which appeared limited to the speB operon.

The investigators are studying an important question, namely how is speB regulated in GAS, an important area of active investigation. The main key finding here is that inactivating RopB increases speB transcript level in a SIP negative background, which indicates that apo-RopB acts as transcriptional repressor of SpeB in contrast to its established role as a key transcriptional activator.

Despite the key strength of the paper noted above, there are several key weaknesses of the paper:

1. The paper is exclusively based on analyses of genetic mutations with no mechanistic insights into the findings
2. The significance of the speB expression identified is not clear because:
 - a. The actual speB expression levels induced by the ropB inactivation relative to wild-type looks very small
 - b. It is not clear that any functional speB is produced (the mature SpeB does not appear to be produced and no functional assays are done)
3. The authors do not do a good job explaining why they found RopB as a speB repressor when all other data that I have seen show it as absolutely required for speB transcription/production

Specific suggestions:

1. The authors change the y axis on many occasions which makes it difficult to discern the actual amount of speB expression. Would suggest trying to keep wild-type as the comparator so that their true magnitude of RopB repression can be better understood.
2. Similarly, when there is both an increase and decrease being shown in the same graph, would use log₂ on the y-axis which helps understand the degree of both better relative to fold-change (like Figure 1F).
3. The finding of a ropB mutant activating speB in Figure 1 is critical to explain better. I have never seen a Western immunoblot for speB where any is detected in a ropB mutant. Did the authors handle their samples differently relative to others (like change pH?) that would have explained this finding? Complementation of their ropB knockout strain to show return to wild-type levels would have been useful given this unexpected finding.
4. The authors should determine if any of their speB expression/speB detection by Western blot translate into functional speB activity. Given the lack of mature speB on their Western blots, it may be that speB activity remains negligible.
5. The authors should not put their model in the Appendix. If a model is needed to help explain their finding, then would include in the main text (although this model is quite confusing).
6. Some type of mechanistic support for their genetic finding would markedly help the manuscript - such as a pull down assay showing altered RopB binding to DNA in the presence and absence of RopB.

Reviewer #3 (Comments to the Authors (Required)):

Chiang-Ni et al set out to provide additional detail on the mechanisms by which RopB regulates expression of SpeB, an important virulence factor of group A Streptococcus. Their methods are adequately described and appropriate and data analysis and presentation is clear. Their conclusions are overall well supported, supporting that RopB free of SIP is not fully without activity. It will be of interest to group A Strep researchers performing work related to RopB and SpeB regulation, that could have further reaching biological importance, though not expanded upon within this study. I have a few minor comments.

An overall comment on mutations, that relates to many experiments: Additional experiments should be done to exclude the possibility of polar effects of RopB/SIP mutation on speB expression, such as 5' RACE or Q-PCR that spans intragenic regions and promoter sites (as depicted in Appendix Fig S4)

Fig 1E. What are the cause for additional intermediate SpeB products in this assay?

Fig 2AB. Why is SpeB so abundant in the covR/ropB mutant, with little expression. In contrast, the covR mutant at 6 h has abundant transcript but little protein. This lies outside the model

Minor:

The introduction should provide detail on what SpeB, as in what does it do during infection and why does it's regulation matter?

The authors should note that CovRS is also referred to as CsrRS (as originally referenced by Wessels, not Scott)

Line 52: the authors should explain what they mean in their interpretation of the Horstmann paper

When discussing negatively regulated by CovR/CovS (as in line 69, but also elsewhere), the authors should clarify if its at a basal state, stimulated, etc

Reviewer #1 (Comments to the Authors (Required)):

Regulation of virulence gene expression in Group A Streptococcus (GAS) is under the control of the master regulator of virulence CovR. This regulation involves the direct and negative regulation of several key virulence genes, but also involves more complex mechanism at specific loci. This manuscript focus on the unusual positive regulation of the SpeB protease by CovR. This regulation involves a regulatory network with a quorum-sensing peptide (SIP), a CovR-regulated protease (PepO), and a SIP-binding activated transcriptional regulator (RopB). In this manuscript, the author show that a specific SIP inactivated mutant is not strictly similar to a RopB deletion mutant for SpeB expression, suggesting that an apo-form of RopB might become a repressor of *speB* transcription.

Major general comments:

It is unfortunate that this manuscript is difficult to read and therefore to evaluate. A recurrent issue is the use of strains, mutants, or genes without prior description. This is partly an editing problem, as some descriptions of mutants are provided in the text after, and not before, the interpretation of experiments. As a result, the overall impression is that of a highly specialized article that is not accessible to readers not directly involved in this specific regulation.

Response:

- (1) The manuscript, including the Abstract and the Result section (paragraphs 1-3), was re-edited (Page 2; Pages 5-9).
- (2) In this revised manuscript, we added the descriptions for the mutants used in the experiments (Lines 97-99, 102, 106, 117, 141-142, 147, 150, 161-162, 168, and 179).

A second major editing issue is the absence of key directly relevant publications in the introduction, especially the Finn et al (mBio 2021,doi.org/10.1128/mBio.01642-21) showing that the expression of *ropB* and *speB* is repressed by nonphosphorylated CovR, and on the high variability of the *ropB* regulon in GAS strains. Despite being cited in the discussion, the Finn et al publication is essential in the introduction.

Response:

The Introduction section was modified according to the reviewer's suggestion (Lines 56-58).

A third editing issue is the difficulties to interpret the experiments due to relative scales often used by the Authors. Usually the relative scales are against a mutant strain without providing first a comparative analysis between the WT and the mutant to infer the level of regulation.

Response:

- (1) In the revised manuscript, we added the qPCR results to demonstrate the relative scales of a mutant strain relative to the wild-type strain in each figure according to the reviewer's suggestion (Fig. 1, Fig. 2, and Fig. 4).
- (2) The expression of *speB* is regulated by multiple regulators, including CovR and RopB described in this study. Therefore, to show the effect of apo-RopB in the regulation of *speB* in different strains, the relative expression level of *speB* in the *ropB* mutant was compared to the wild-type strain, the *covR* mutant, and the *covS* mutants in Fig. 1 – Fig. 3. To make readers

easier to follow, the reference gene utilized for relative quantification in each figure was indicated by the inverted triangle in this revised manuscript (Fig. 1 – Fig. 5).

It should be also noted that no genetic complementation of mutant phenotypes is provided in the manuscript. Especially, it is unclear if the *SIP** mutant does not have secondary mutations in its genome, or if the introduced mutation does not impact the transcription of the operon, the stability of the RNA, or the translational efficiency of the other genes.

Response:

The *SIP trans*-complemented strain was included in Figure 1B. Also, the *ropB trans*-complementary strain was included in Figure 1G.

Comments:

Abstract: I can only suggest to re-write the abstract. The abstract is difficult to read without prior knowledge of the regulatory systems and their components and of the specific literature. For instance, genes are ill-defined. In addition, it is somehow unclear what have been previously published and what is the novelty.

Response:

The abstract was re-written according to the reviewer's comments (Page 2).

Introduction line 75: I don't understand what "has not been inactivated but transformed" means.

Response:

The sentence was deleted and the Introduction section was extensively modified (Lines 84-89).

Results, lines 82-96: please defined the strain used and the type of inactivating mutation for the *SIP** mutant. For Fig. 1 A-D, please provided first the WT controls and included the single mutants (*pepO*, *ropB*, ...) . An option will be to first described the actual panel E. It will be also useful to describe for non-specialists what are the zymogen and mature forms of *SpeB* (*zSpeB*, and *mSpeB*). It is unclear what 'if the regulatory activity of *RopB* is inactivated in the absence of *SIP*' and 'would have an undefined role' means. Please rephrase them.

Responses:

- (1) The description for the *SIP** mutant was added in the first paragraph of the Results section (Lines 97-99).
- (2) The wild-type strain control, the *pepO* isogenic mutant, and the *ropB* mutant were included in Figure 1A and Figure 1G.
- (3) The introduction for *SpeB* (*zSpeB* and *mSpeB*) was added in the Introduction section (Lines 63-65).
- (4) The sentence ('if the regulatory activity of *RopB* is inactivated in the absence of *SIP*' and 'would have an undefined role') was deleted and this paragraph was extensively modified (Lines 114-121).

Results, lines 97-108 (corresponding to fig. 1): The results corresponding to Fig.1 are difficult to

follow. The main results are: first a confirmation that PepO degrades SIP (natural and synthetic), and second that there is a difference of between *ropB* and *sip** mutants for *SpeB* expression. The conclusion is that *RopB* could act as a transcriptional repressor of *speB* in the absence of SIP. I am not convince since other hypothesis can be made at this stage (indirect regulation, *CovR* regulation). The Authors should be more cautious in their hypotheses and experimental interpretation.

Responses:

- (1) The first paragraph of the Results section was extensively modified (Lines 92-134).
- (2) In agreed with the reviewer's comment, in this paragraph, we cannot exclude the role of *CovR* in regulation of *speB*. Therefore, the role of *RopB* in the regulation of *speB* in the *covR* mutant was analyzed in the second paragraph of the Results section (Fig. 2).

Results, lines 110-129 (corresponding to fig. 2): It will really help readers (and the reviewer) to have a wild-type control in fig. 2. It is also unclear how the overexpression of *PepO* in a Δ *covR* mutant has an effect in these conditions. It is unfortunate that *pepO* mutant where studies in fig. 1 and 3 but not in Fig. 2 where its effect is most likely the strongest.

Responses:

- (1) The wild-type strain control was included (Fig. 2A) according to the reviewer's suggestion.
- (2) The role of *PepO* in the regulation of *speB* in the *covR* mutant has been described in our publication (Microbiol Spectr 2022: <https://doi.org/10.1128/spectrum.02033-22>). According to the reviewer's suggestion, we also presented the results from the *pepO* mutant and Δ *covR*/ Δ *pepO* mutant in Fig. 2A.

Results, lines 131-157 (corresponding to fig. 3): Please define the *covS* mutant (line 133) before interpreting the result. Results obtained with different level of *CovR* phosphorylation are interesting. However, it is unfortunate that a *CovR* D53A mutant, a non phosphorylable variant of *CovR*, not included in this study. This will be interesting, especially considering that the non-phosphorylated form of *CovR* appears to bind and regulate the *speB* and *sip* promoters (Finn et al 2021).

Responses:

- (1) The definition of the *covS* mutant was added in the text according to the reviewer's suggestion (Lines 161-162).
- (2) The *CovR* D53A mutant was included in Fig. 3G according to the reviewer's comments (Lines 188-194).

Results, lines 159-181 (corresponding to fig. 4): Once again, results are difficult to read and evaluate (double mutant versus WT and double mutant versus single mutant). It seems to me that the transcriptome, or at least qRT-PCR, of the single *rpoB* mutant is necessary to sustain the conclusion. Currently, it appear that the *SIP** mutant specifically impact the transcription of the *sip*-containing operon, raising concern on the stability of the RNA due to the mutation. In the absence of confirmatory experiments (maybe by exogenous complementation with synthetic or purified SIP), it is difficult to be convinced by the specific transcriptional regulation proposed by the Author.

Responses:

- (1) The paragraph was modified (Lines 201-204).
- (2) The *ropB* isogenic mutant was included in this figure (Fig. 4D) according to the reviewer's suggestion (Lines 214-223).
- (3) The *SIP trans*-complementary strain was included in this revised manuscript (Fig. 1B). Results supported that complementation of SIP in the *SIP** mutant could restore the expression of SpeB in both transcriptional and translational levels. Also, the exogenous supplementation of synthetic SIP upregulated the SpeB expression in the *SIP** mutant (Fig. 1C). These results suggested that the *speB* transcription is not impaired by undefined factors.

Unfortunately, the direct binding of RpoB (Apo and with SIP) on the promoter has not been tested. This is a critical experiment that is necessary to not sustain the conclusion on transcriptional data only. The characterization of different binding sites for the Apo and the complexed RpoB will give a significant strength to the hypothesis of an activator becoming a repressor. The Authors should also considered (and tested) alternative hypothesis such as a competitive binding between CovR and RpoB.

Responses:

- (1) The direct binding of RpoB (Apo and with SIP) on the *speB* promoter has been tested before (Do et al. 2017. <https://doi.org/10.1073/pnas.1705972114>, Supplementary Fig. S6) and the results suggested that apo-RopB and RopB-SIP have similar binding activity to the *speB* promoter.
- (2)) We also repeated this experiment. The results were similar to the published data from Do et al (2017) and were shown in below.

[Figure removed by LSA Editorial Staff per authors' request]

- (3) We utilized the *speB* promoter-deletion mutants to elucidate which region could be essential for the apo-RopB-mediated *speB* repression (Supplementary Fig. S4). Results from the previous studied and our study both showed that the full length of *speB* promoter was required for *speB* expression; therefore, the mechanism of apo-RopB-mediated *speB* repression cannot be further verified. Unfortunately, in the current stage, we cannot provide further mechanistic insights about how apo-RopB represses *speB* transcription.
- (4) We confirmed that the apo-RopB could repress *speB* expression in the *covR* mutant (Fig. 2); therefore, the repression of *speB* by the apo-RopB would be CovR-independent.

Reviewer #2 (Comments to the Authors (Required)):

RopB represses the transcription of *speB* in the absence of SIP in group A streptococcus

In this article, the authors investigate the transcription/production of the key cysteine protease SpeB. They use a mutant strain of GAS in which the SpeB inducing peptide (SIP) has been inactivated. It has previously been shown that RopB-SIP interaction under acidic conditions is critical to *speB* expression. In figure 1 they show that inactivating RopB in the background of SIP inactivation increases *speB* expression and production. Figure 2 shows similar findings in a CovR background. Figure 3 introduces the endopeptidase pepO into *speB* regulation and shows differential impact of *ropB* inactivation dependent on CovS activity. The remainder of the manuscript focuses on the genome-wide impact of SIP which appeared limited to the *speB* operon.

The investigators are studying an important question, namely how is *speB* regulated in GAS, an important area of active investigation. The main key finding here is that inactivating RopB increases *speB* transcript level in a SIP negative background, which indicates that apo-RopB acts as transcriptional repressor of SpeB in contrast to its established role as a key transcriptional activator.

Despite the key strength of the paper noted above, there are several key weaknesses of the paper:

1. The paper is exclusively based on analyses of genetic mutations with no mechanistic insights into the findings

Responses:

(1) The previous study has shown that apo-RopB and RopB-SIP have similar binding activity to *speB* promoter (Do et al. 2017. <https://doi.org/10.1073/pnas.1705972114>, (Supplementary Fig. S6).

We also repeated the electrophoresis mobility shift assay, and the results were similar to the data from Do et al (2017) (as shown in below).

[Figure removed by LSA Editorial Staff per authors' request]

(2) Also, we utilized the *speB* promoter-deletion mutants to elucidate which region could be essential for the apo-RopB-mediated *speB* repression (Supplementary Fig. S4). Results from the previous studied and our study both showed that the full length of *speB* promoter was required for *speB* expression; therefore, the mechanism of apo-RopB-mediated *speB* repression cannot be further verified. Unfortunately, in the current stage, we cannot provide further mechanistic insights about how apo-RopB represses *speB* transcription.

- (3) Although we did not reveal the mechanism of how apo-RopB represses *speB* transcription, this study reveals that the *speB* repression in the *covS* mutant is mediated by apo-RopB. This finding provides a novel insight into the interaction between CovR/CovS and RopB-SIP systems.

2. The significance of the *speB* expression identified is not clear because:

- a. The actual *speB* expression levels induced by the *ropB* inactivation relative to wild-type looks very small

Response:

RopB is required to activate the *speB* expression in the stationary phase of growth and under the acidic conditions (Do et al. 2019. <https://doi.org/10.1038/s41467-019-10556-8>; Neely et al. 2003. <https://dx.doi.org/10.1128%2FJ.B.185.17.5166-5174.2003>). Nonetheless, in the exponential phase or under the neutral conditions, RopB cannot trigger the *speB* expression (Neely et al. 2003. <https://dx.doi.org/10.1128%2FJ.B.185.17.5166-5174.2003>); why RopB cannot trigger *speB* transcription in certain conditions was not clear. As indicated by the reviewer, the role of apo-RopB in the regulation of *speB* expression is relatively minor compared to the RopB-SIP. Nonetheless, we first revealed that apo-RopB would act as the transcriptional repressor of *speB*, and this finding is important for understanding how RopB regulates the *speB* expression under different conditions.

- b. It is not clear that any functional *speB* is produced (the mature SpeB does not appear to be produced and no functional assays are done)

Response:

- (1) The Introduction section was modified to include more information about SpeB (Lines 63-65).

- (2) The SpeB protease activity is essential to catalyze the zymogen form SpeB (42 kDa) to the mature form protease (28 kDa). As shown in the previous study, the inactivation of SpeB protease activity by mutation of its catalytic residues (Cys192 or His340) abolish the SpeB maturation (Chen et al. 2003. <https://doi.org/10.1074/jbc.M209038200>, Fig. 2).

We also repeated the experiment (SpeB_{C192S}, the cysteine protease-inactivated mutant), and results were showed in below:

[Figure removed by LSA Editorial Staff per authors' request]

In conclusion, the 28 kDa mature form SpeB can be detected by the western blot hybridization, indicating that the GAS strain can produce SpeB and the SpeB has protease activity.

3. The authors do not do a good job explaining why they found RopB as a *speB* repressor when all other data that I have seen show it as absolutely required for *speB* transcription/production

Responses:

- (1) We defined apo-RopB as a *speB* repressor based on comparing the *speB* expression in the SIP-inactivated mutant (*SIP**) and its isogenic *ropB* mutant (*SIP*/ΔropB*). The apo-RopB-

mediated *speB* repression was observed in the wild-type strain (Fig. 1), the *covR* mutant (Fig. 2), and the *covS* mutant (Fig. 3); therefore, we concluded that apo-RopB would act as the transcriptional repressor of *speB*.

- (2) As indicated by the reviewer, we agreed that the role of apo-RopB in the regulation of *speB* expression is relatively minor compared to the RopB-SIP; however, apo-RopB-mediate *speB* repression was not reported previously and could be crucial for us to understand how *speB* is regulated by RopB under different conditions.

Specific suggestions:

1. The authors change the y axis on many occasions which makes it difficult to discern the actual amount of *speB* expression. Would suggest trying to keep wild-type as the comparator so that they true magnitude of RopB repression can be better understood.

Response:

- (1) In the revised manuscript, we added the qPCR results to demonstrate the relative scales of a mutant strain relative to the wild-type strain in each figure according to the reviewer's suggestions (Fig. 1, Fig. 2, and Fig. 4).
- (2) The expression of *speB* is regulated by multiple regulators, including CovR and RopB described in this study. Therefore, to show the effect of apo-RopB in the regulation of *speB*, the relative expression level of *speB* in the *ropB* mutant was compared to the wild-type strain, the *covR* mutant, and the *covS* mutants in different figures. To make readers easier to follow, the reference gene utilized for relative quantification in each figure was indicated by the inverted triangle in this revised manuscript (Fig. 1 – Fig. 5).

2. Similarly, when there is both an increase and decrease being shown in the same graph, would use log₂ on the y-axis which helps understand the degree of both better relative to fold-change (like Figure 1F).

Response:

- (1) As indicated by the reviewer, log₂ on the y-axis would clearly show the relative fold change. Nonetheless, to the overall consistency of the data presentation, we did not change the y-axis presentation in this revised manuscript.
- (2) As described in the previous Response, the reference gene utilized for relative quantification in each figure was indicated by the inverted triangle and we hope this change could make readers easier to follow the data presented in this revised manuscript (Fig. 1 – Fig. 5).

3. The finding of a *ropB* mutant activating *speB* in Figure 1 is critical to explain better. I have never seen a Western immunoblot for *speB* where any is detected in a *ropB* mutant. Did the authors handle their samples differently relative to others (like change pH?) that would have explained this finding? Complementing their *ropB* knockout strain to show return to wild-type levels would have been useful given this unexpected finding.

Responses:

- (1) The signal intensity from the western blot hybridization is based on the exposure time. The wild-type strain expresses a large amount of SpeB in the stationary phase; therefore, the SpeB

signal from the *ropB* mutant is relatively weak and usually ignored if the exposure time is short.

- (2) According to the reviewer's suggestion, the expression of SpeB in the *ropB trans*-complementation strain was shown in Fig. 1G.

4. The authors should determine if any of their speB expression/speB detection by Western blot translate into functional speB activity. Given the lack of mature speB on their Western blots, it may be that speB activity remains negligible.

Response:

Please refer to the Response for the comment 2-b.

5. The authors should not put their model in the Appendix. If a model is needed to help explain their finding, then would include in the main text (although this model is quite confusing).

Response:

The manuscript has been edited and the model figure was moved to Fig. 6 according to the reviewer's suggestions. Further, we also edited the model figure to make it easier for reading.

6. Some type of mechanistic support for their genetic finding would markedly help the manuscript - such as a pull down assay showing altered RopB binding to DNA in the presence and absence of RopB.

Response:

Please refer to the Responses for the reviewer's comment 1-(1).

Reviewer #3 (Comments to the Authors (Required)):

Chiang-Ni et al set out to provide additional detail on the mechanisms by which RopB regulates expression of SpeB, an important virulence factor of group A Streptococcus. Their methods are adequately described and appropriate and data analysis and presentation is clear. Their conclusions are overall well supported, supporting that RopB free of SIP is not fully without activity. It will be of interest to group A Strep researchers performing work related to RopB and SpeB regulation, that could have further reaching biological importance, though not expanded upon within this study. I have a few minor comments.

An overall comment on mutations, that relates to many experiments: Additional experiments should be done to exclude the possibility of polar effects of RopB/SIP mutation on *speB* expression, such as 5' RACE or Q-PCR that spans intragenic regions and promoter sites (as depicted in Appendix Fig S4)

Response:

The *SIP trans*-complementary strain was included in this revised manuscript (Fig. 1B). Results supported that complementation of SIP in the *SIP** mutant could restore the expression of SpeB in both transcriptional and translational levels. Also, the exogenous supplementation of synthetic SIP upregulated the SpeB expression in the *SIP** mutant (Fig. 1C). These results suggested that the *speB* transcription is not impaired by undefined factors.

Fig 1E. What are the cause for additional intermediate SpeB products in this assay?

Response:

The SpeB cysteine protease is secreted to the medium as the zymogen form (42 kDa) that is autocatalyzed to the mature protease (28 kDa) (Doran et al. 1999. <https://doi.org/10.1046/j.1432-1327.1999.00473.x>). During the autocatalytic process, the intermediate SpeB products could be detected by western blot hybridization. This information was included in the Introduction section of this revised manuscript (Lines 63-65).

Fig 2AB. Why is SpeB so abundant in the *covR/ropB* mutant, with little expression. In contrast, the *covR* mutant at 6 h has abundant transcript but little protein. This lies outside the model

Response:

The transcription of *pepO*, *ropB*, and *speB* are repressed by CovR. Based on our model, the upregulation of *pepO* would degrade the quorum-sensing peptide SIP; therefore, in the exponential phase (after 5 h of incubation), the apo-RopB would repress *speB* transcription. In the stationary phase, the upregulation of *ropB* and *speB* would compensate for the effect of PepO degradation and therefore the *speB* transcription was increased dramatically (after 6-7 h of incubation). The *speB* transcription in the *covR* mutant was inhibited by apo-RopB in the exponential phase of growth; therefore, the SpeB protein started to translate after 6 h of incubation and a large amount of SpeB protein was detected after 7 h of incubation. In contrast, the transcription was *speB* was not inhibited by apo-RopB in the $\Delta covR/\Delta ropB$ mutant. To compare with the *covR* mutant, the $\Delta covR/\Delta ropB$

mutant expressed a higher level of *speB* after 5 h of incubation. Therefore, in the $\Delta covR/\Delta ropB$ mutant, SpeB can be detected after 5 h of incubation.

Minor:

The introduction should provide detail on what SpeB, as in what does it do during infection and why does its regulation matter?

Response:

The Introduction section was modified according to the reviewer's comments (Lines 65-68).

The authors should note that CovRS is also referred to as CsrRS (as originally referenced by Wessels, not Scott)

Response:

The text was modified according to the reviewer's comments (Line 46).

Line 52: the authors should explain what they mean in their interpretation of the Horstmann paper

Response:

According to other reviewer's suggestions, the Introduction section was extensively modified (Lines 56-62) and the sentence indicated by the reviewer was deleted.

When discussing negatively regulated by CovR/CovS (as in line 69, but also elsewhere), the authors should clarify if its at a basal state, stimulated, etc

Response:

In this manuscript, we only discussed the regulation of CovR/CovS under the regular culture condition (without additional stimulations).

March 6, 2023

Re: Life Science Alliance manuscript #LSA-2022-01809-TR

Prof. Chuan Chiang-Ni
Chang Gung University
259 Wen-Hwa 1st Road, Kwei-Shan
Taoyuan 33323
Taiwan

Dear Dr. Chiang-Ni,

Thank you for submitting your revised manuscript entitled "RopB represses the transcription of speB in the absence of SIP in group A Streptococcus" to Life Science Alliance. The manuscript has been seen by the original reviewers whose comments are appended below. While the reviewers continue to be overall positive about the work in terms of its suitability for Life Science Alliance, some important issues remain. As raised by two reviewers the manuscript needs to be professionally edited, including the abstract. Also, as pointed out by Reviewer 1, the RNA-seq data needs to be better explained and since the overall observed effects are minor, the authors need to deem down their conclusions throughout the manuscript.

Our general policy is that papers are considered through only one revision cycle; however, given that the suggested changes are relatively minor, we are open to one additional short round of revision. Please note that I will expect to make a final decision without additional reviewer input upon resubmission.

Please submit the final revision within one month, along with a letter that includes a point by point response to the remaining reviewer comments.

To upload the revised version of your manuscript, please log in to your account: <https://lsa.msubmit.net/cgi-bin/main.plex>
You will be guided to complete the submission of your revised manuscript and to fill in all necessary information.

B. MANUSCRIPT ORGANIZATION AND FORMATTING:

Sincerely,

Reviewer #1 (Comments to the Authors (Required)):

I apologize but I am unable to properly evaluate this revised manuscript. Although the authors have significantly edited the text, it remains extremely difficult to read and evaluate. For example, the revised abstract is simply not understandable. I can only suggest that authors have their manuscript professionally edited or edited by a native English-speaking scientist.

The responses to the initial comments and suggestions are very brief, and not convincing. Results are still difficult to follow and the model (Fig. 6) confusing.

Overall, the observed effects are minor, and the authors did not convincingly demonstrate their hypothesis of an activator turning into a repressor. No mechanism has been explored and an indirect effect cannot be excluded.

Additional comment.

RNA-seq (Fig. 4A, 4B, S1) are still very confusing. If I have understood the graphs correctly, a large group of genes are down-regulated on such a massive scale ($\log_2 > 20$ fold !!!!) that this can only correspond to a large deletion in the ropB mutant. And this includes the rgg regulator!! The authors did not even mention this group of genes in the result section (line 196-218), nor the massive overexpression of spy1733 and the discrepancy between RNA-seq and RT-qPCR value.

Reviewer #2 (Comments to the Authors (Required)):

The authors have improved and revised their manuscript in accordance with reviewers' suggestions. The abstract remains in need in of improvement but I do not have additional suggestions at the present.

Reviewer #3 (Comments to the Authors (Required)):

1. Chiang-Ni and co-authors show that in the the RopB regulator not only activates speB transcription when bound to SIP, but represses in the absence of SIP.
2. In their revised manuscript, they have satisfactorily addressed my technical concerns and show data supporting their claims.
3. The revised manuscript is written more clearly, though figure legends contain some detail better suited for methods. The statistics are likely inaccurate in places, as broadly stated to all be 'Tukey', which would not be valid, or even possible, for some of these. There is also significant chart clutter with comparisons not discussed that may be meaningless.

Reviewer #1 (Comments to the Authors (Required)):

I apologize but I am unable to properly evaluate this revised manuscript. Although the authors have significantly edited the text, it remains extremely difficult to read and evaluate. For example, the revised abstract is simply not understandable. I can only suggest that authors have their manuscript professionally edited or edited by a native English-speaking scientist.

Responses:

The abstract was modified, and the manuscript was edited by professional English editing.

The responses to the initial comments and suggestions are very brief, and not convincing. Results are still difficult to follow and the model (Fig. 6) confusing.

Responses:

We are trying to respond to the reviewer's comments as concisely as possible. The manuscript has been edited by a professional editing service, and the description of the model (Fig. 6) was modified (Lines 253-255 and the figure legend of Fig. 6).

Overall, the observed effects are minor, and the authors did not convincingly demonstrate their hypothesis of an activator turning into a repressor. No mechanism has been explored and an indirect effect cannot be excluded.

Responses:

- (1) These comments were raised by reviewer-1 and reviewer-2. In the first revision cycle, the reviewer-2 has accepted our responses.
- (2) We agreed that the role of SIP-free RopB (apo-RopB) in the regulation of *speB* expression is relatively minor compared to the SIP-bound RopB (RopB-SIP); however, apo-RopB-mediated *speB* repression was not reported previously and could be crucial for us to understand how *speB* is regulated by RopB under different conditions.
- (3) We demonstrated that the apo-RopB represses the *speB* expression in the wild-type strain (Fig. 1), in the *covR* mutant (Fig. 2D-2E), and in the *covS* mutant (Fig. 3E-3G); therefore, we concluded that apo-RopB would act as the transcriptional repressor of *speB*.
- (4) In the first revision cycle, we provided the EMSA analysis results to demonstrate apo-RopB could bind to *speB* promoter DNA. This result is consistent with the study from Do et al. 2017. (<https://doi.org/10.1073/pnas.1705972114>), suggesting that apo-RopB could regulate *speB* expression directly. We cannot completely exclude the possibility that there could be unknown factors to affect *speB* transcription indirectly. Nonetheless, the current study showed that the whole *speB* promoter region is required for *speB* transcription (Fig. S4); therefore, the mechanism of apo-RopB-mediated *speB* repression cannot be further verified.

Additional comment.

RNA-seq (Fig. 4A, 4B, S1) are still very confusing. If I have understood the graphs correctly, a large group of genes are down-regulated on such a massive scale ($\log_2 > 20$ fold !!!!) that this can only correspond to a large deletion in the *ropB* mutant. And this includes the *rgg* regulator!! The authors did not even mention this group of genes in the result section (line 196-218), nor the massive

overexpression of *spy1733* and the discrepancy between RNA-seq and RT-qPCR value.

Responses:

- (1) The upregulated and down-regulated genes in the *ropB* mutant, including *Spy1176*, *Spy1416*, and *Spy1426*, were described in Lines 218-229 and the analyzed results were shown in Fig. 5 and Fig. S2.
- (2) RNA sequence is used to compare the abundance of RNA transcripts in different strains. The *SIP*/ Δ ropB* mutant is the *ropB* gene deleted strain; therefore, in Fig. 4A and Fig. 4B, to compare with the wild-type strain and the *SIP** mutant (both have the *ropB* gene and transcribe the *ropB* RNA), the transcription of *ropB* is downregulated in the *SIP*/ Δ ropB* mutant.
- (3) *Spy1733* is the hypothetical open reading frame and was annotated differently in MGAS5005 and SF730 (described in Lines 198-202). In RNA sequencing analysis, the transcripts of *Spy1733*, including *speB-spi-spy1733* co-transcript and *Spy1733* itself (*prsA*, as shown in Fig. 4C), were detected. Nonetheless, in the RT-qPCR analysis, we utilized primers to detect *speB-spi-spy1733* co-transcript and *prsA* separately (Fig. 4C). Therefore, although RNA sequence and RT-qPCR analyses both showed the expression of *Spy1733* in the *SIP*/ Δ ropB* mutant was upregulated when compared to the *SIP** mutant, the level of *Spy1733* upregulation is not completely identical in RNA sequence and RT-qPCR analyses.

Reviewer #2 (Comments to the Authors (Required)):

The authors have improved and revised their manuscript in accordance with reviewers' suggestions. The abstract remains in need in of improvement but I do not have additional suggestions at the present.

Response:

The abstract was modified, and the manuscript was edited by professional English editing.

Reviewer #3 (Comments to the Authors (Required)):

1. Chiang-Ni and co-authors show that in the the RopB regulator not only activates *speB* transcription when bound to SIP, but represses in the absence of SIP.
2. In their revised manuscript, they have satisfactorily addressed my technical concerns and show data supporting their claims.
3. The revised manuscript is written more clearly, though figure legends contain some detail better suited for methods. The statistics are likely inaccurate in places, as broadly stated to all be 'Tukey', which would not be valid, or even possible, for some of these. There is also significant chart clutter with comparisons not discussed that may be meaningless.

Response:

- (1) The figure legends (including the figure legend of Fig. 1, Fig. 3, and Fig. S2) were modified according to the reviewer's suggestion.
- (2) The statistics (in the Materials and methods section; Lines 402-409) was modified. The statistical methods for RNA sequencing analyses were included in this revised manuscript.
- (3) According to the reviewer's comments, the chart clutters in Fig. 1C (*SIP*/ Δ ropB*, SIP 0, 0.1, and 0.5 μ M) and Fig. 1H (6h and 7 h) were removed.

March 20, 2023

RE: Life Science Alliance Manuscript #LSA-2022-01809-TRR

Prof. Chuan Chiang-Ni
Chang Gung University
259 Wen-Hwa 1st Road, Kwei-Shan
Taoyuan 33323
Taiwan

Dear Dr. Chiang-Ni,

Thank you for submitting your revised manuscript entitled "RopB represses the transcription of speB in the absence of SIP in group A Streptococcus". We would be happy to publish your paper in Life Science Alliance pending final revisions necessary to meet our formatting guidelines.

- please add the Twitter handle of your host institute/organization as well as your own or/and one of the authors in our system
- please add the author contributions to the main manuscript text
- please upload your table files as editable doc or excel files or make sure that they are in the doc file of your main manuscript text

Figure Check:

- please add sizes next to all blots

A. FINAL FILES:

B. MANUSCRIPT ORGANIZATION AND FORMATTING:

Sincerely,

March 23, 2023

RE: Life Science Alliance Manuscript #LSA-2022-01809-TRRR

Prof. Chuan Chiang-Ni
Chang Gung University
259 Wen-Hwa 1st Road, Kwei-Shan
Taoyuan 33323
Taiwan

Dear Dr. Chiang-Ni,

Thank you for submitting your Research Article entitled "RopB represses the transcription of speB in the absence of SIP in group A Streptococcus". It is a pleasure to let you know that your manuscript is now accepted for publication in Life Science Alliance. Congratulations on this interesting work.

DISTRIBUTION OF MATERIALS:

Again, congratulations on a very nice paper. I hope you found the review process to be constructive and are pleased with how the manuscript was handled editorially. We look forward to future exciting submissions from your lab.

Sincerely,
